

# Technical note: The CAMS greenhouse gas reanalysis from 2003 to 2020

Anna Agustí-Panareda[1], Jérôme Barré[1], Sébastien Massart[1], Antje Inness[1], Ilse Aben[2], Melanie Ades[1], Bianca C. Baier[3,4], Gianpaolo Balsamo[1], Tobias Borsdorff [2], Nicolas Bousserez[1], Souhail Boussetta[1], Michael Buchwitz[5], Luca Cantarello[1], Cyril Crevoisier[6], Richard Engelen[1], Henk Eskes[7], Johannes Flemming[1], Sébastien Garrigues[1], Otto Hasekamp[2], Vincent Huijnen[7], Luke Jones[1], Zak Kipling[1], Bavo Langerock[8], Joe McNorton[1], Nicolas Meilhac[6], Stefan Noel[5], Mark Parrington[1], Vincent-Henri Peuch[1], Michel Ramonet[9], Miha Ratzinger[1], Maximilian Reuter[5], Roberto Ribas[1], Martin Suttie[1], Colm Sweeney[4], Jérôme Tarniewicz[9], Lianghai Wu[10]

[1] European Centre for Medium Range Weather Forecasts, Shenfield Park, Reading RG2 9AX, United Kingdom
[2] SRON Netherlands Institute for Space Research, Utrecht, the Netherlands
[3] Cooperative Institute for Research in Environmental Sciences, University of Colorado-Boulder, Boulder, CO, USA
[4] NOAA, Global Monitoring Laboratory, Boulder, CO, USA
[5] Institute of Environmental Physics (IUP), University of Bremen, 28334 Bremen, Germany
[6] Laboratoire de Météorologie Dynamique (LMD/IPSL), CNRS, Ecole polytechnique, 91128 Palaiseau Cedex, France
[7] Royal Netherlands Meteorological Institute, Utrechtseweg 297, NL-3731 GA De Bilt, Netherlands
[8] Royal Belgian Institute for Space Aeronomy, Avenue Circulaire 3, 1180 Uccle, Belgium
[9] Laboratoire des Sciences du Climat et de l'Environnement (LSCE-IPSL), CEA-CNRS-UVSQ, Université Paris-Saclay, 91191 Gif-sur-Yvette, France
[10] Flemish Institute for Technological Research (VITO), Remote Sensing Unit, Boeretang 200, B-2400 Mol, Belgium

*Correspondence to*: Anna Agusti-Panareda (A.Agusti-Panareda@ecmwf.int)

**Abstract.** The Copernicus Atmosphere Monitoring Service has recently produced a greenhouse gases reanalysis (version egg4) that covers almost two decades from 2003 to 2020 and will be extended in the future. This reanalysis dataset includes carbon dioxide ($CO_2$) and methane ($CH_4$). The reanalysis procedure combines model data with satellite data into a globally complete and consistent dataset using the European Centre for Medium-range Weather Forecasts' Integrated Forecasting System (IFS). This dataset has been carefully evaluated against independent observations to ensure validity and point out deficiencies to the user. The greenhouse gas reanalysis can be used to examine the impact of atmospheric greenhouse gases concentrations on climate change, such as global and regional climate radiative forcing, assess intercontinental transport, and also serve as boundary conditions for regional simulations, among other applications and scientific studies. The caveats associated with changes in assimilated observations and fixed underlying emissions are highlighted, as well as their impact on the estimation of trends and annual growth rates of these long-lived greenhouse gases.



## 1 Introduction

Atmospheric carbon dioxide ($CO_2$) and methane ($CH_4$) are the most abundant man-made greenhouse gases directly responsible for climate change (IPCC, 2021). Their long lifetime and increasing anthropogenic emissions near the surface account for their long-term trends (Friedlingstein et al., 2021). A lot of effort has been devoted to measuring the atmospheric concentrations from ground-based observatories (e.g. National Oceanic and Atmospheric Administration (NOAA), gml.noaa.gov; Integrated

Carbon Observation System (ICOS), www.icos-cp.eu), which provide the gold standard for the estimation of trends, and more recently satellite data (Committee on Earth Observation Satellites (CEOS), Crisp et al., 2018) enhancing the spatial coverage of greenhouse gas observations at global scale. Atmospheric measurements also sample the variability of $CO_2$ and $CH_4$ coming from the weather and its associated atmospheric transport (e.g. Patra et al., 2008, 2011). For this reason, Numerical Weather Prediction (NWP) models have been extensively used to represent and reconstruct the variability of atmospheric concentrations

of various tracers (e.g. Inness et al., 2019). Here we use the Integrated Forecasting System (IFS) of the European Centre for Medium-range Weather Forecasts (ECMWF) which has been adapted to include $CO_2$ and $CH_4$ in the weather forecast (Agustí-Panareda et al., 2017, 2019) to create a greenhouse gases (GHG) reanalysis. The reanalysis uses the data assimilation technique to combine $CO_2$ and $CH_4$ satellite data from the SCanning Imaging Absorption spectroMeter for Atmospheric CHartographY (SCIAMACHY, www.sciamachy.org ), the Infrared Atmospheric Sounding Interferometer (IASI, www.eumetsat.int/iasi) and

The Thermal and Near Infrared Sensor for Carbon Observation (TANSO, www.eorc.jaxa.jp/GOSAT/instrument_1.html) instruments with IFS model simulations of $CO_2$ and $CH_4$ (Agustí-Panareda et al., 2022). The dataset is based on a consistent and stable model version to provide a homogenous, continuous and gapless record of the $CO_2$ and $CH_4$ in the entire atmosphere since 2003.

The IFS includes a forecasting model and a data assimilation system combined. The data assimilation system also integrates

meteorological observations as in the fifth generation of ECMWF meteorological reanalyses, ERA5 (Hersbach et al., 2020), to best constrain the atmospheric variability of greenhouse gases (Massart et al., 2014, 2016). The forecasting model provides a 3-dimensional representation and evolution of the atmospheric $CO_2$ and $CH_4$ and meteorological variables (Agustí-Panareda et al., 2019). At the model surface the greenhouse gases are forced by a set of surface fluxes and emissions. Such modelling configuration allows to produce a realistic representation of the spatio-temporal variability of greenhouse gases in the

atmosphere over a wide range of scales from hours to seasons and from local to global (Agustí-Panareda et al., 2022).

Figure 1 showcases the global evolution of $CO_2$ and $CH_4$ represented by the CAMS GHG reanalysis data set over the period 2003-2020 and the span of the used satellite data. The seasonal averages illustrate the spatial and temporal variability information contained in the reanalysis dataset which can be exploited for a range of applications in atmospheric sciences. A

key potential use of the CAMS GHG reanalysis is to assess the impact of greenhouse gases on climate change. The reanalysis 3-dimensional fields could be used to investigate global and regional climate radiative forcing (e.g. atmosphere.copernicus.eu/climate-forcing), serve as boundary conditions for regional simulations, assess intercontinental





transport, and generally provide a reference for any other study focusing on atmospheric variability of $CO_2$ and $CH_4$. However, care should be taken when using the CAMS GHG reanalysis to estimate trends and annual growth rates of these long-lived

greenhouse gases by considering the caveats associated with the changes in the satellite retrievals of $CO_2$ and $CH_4$ and the fact that neither anthropogenic emissions nor natural fluxes are adjusted by the data assimilation system, unlike atmospheric inversions (e.g. Chevallier et al., 2019).

The objective of this technical report is to document the technical aspects of the method and input data used to produce the CAMS GHG reanalysis, and to provide guidance to potential users on the strengths and limitations of the dataset. Section 2

describes the processing chain to produce the reanalysis and its components. Section 3 focuses on the evaluation of the CAMS GHG reanalysis using independent observations from the TCCON and NDACC networks as well as surface in situ networks and AirCore profiles. A list of limitations and caveats of the CAMS GHG reanalysis associated with the changes in the assimilated data and the underlying model errors are compiled in Sect. 4. Finally, Sect. 5 provides a summary and outlook for future CAMS GHG reanalyses.





**Figure 1.** (a) Reanalysis timeseries of global column-averaged $CO_2$ (red) and $CH_4$ (purple) atmospheric mole fractions; (b) the span of the satellite data records for the corresponding species; (c) $CO_2$ and (d) $CH_4$ seasonal total column averages (DJF: December-January-February, MAM: March-April-May, JJA: June-July-August, SON: September, October, November) for the 2003-2020 period.




**Figure 2. Schematic of the reanalysis cycling procedure. The flow diagram shows the steps and elements combined in the reanalysis. Surface fluxes are used as boundary condition for the atmospheric forecasts. Satellite data are combined with the forecast using data assimilation to produce an analysis (corrected 4D fields) to initialize the next forecast.**






## 2 Methods

This section gives an overview of the different building blocks of the CAMS GHG reanalysis and the processing chain that integrates the different components to produce the reanalysis dataset.

### 2.1 The reanalysis cycling chain

The reanalysis production chain is illustrated in Fig. 2. It is a cycling procedure based on a 12-hour data assimilation window that involves four main parts:

- Satellite retrievals of $CO_2$ and $CH_4$ (see section 2.2) as well as NWP observations (Hersbach et al., 2020).
- Surface fluxes (see section 2.3) that constitute the sources and sinks of $CO_2$ and $CH_4$ in the atmosphere are compiled from various sources. They provide the surface boundary condition for the tracer transport model.
- A model forecast (see section 2.4) that provides a 4-dimentional representation of the state of the greenhouse gases over space and time, along with other meteorological variables, during the 12-hour analysis window (from 09:00 to 21:00 and 21:00 to 09:00 UTC). The forecasts are initialised with the previous analysis, except for the first forecast for the initial date, which is initialised with atmospheric molar fractions from the CAMS inversion dataset (Chevallier et al., 2020; Segers et al., 2020a).
- The above elements are combined using a data assimilation system (see section 2.5) to produce an analysis (Massart et al., 2014, 2016). The analysis will serve to initialise the following forecast over the subsequent 12 hourly cycle.

Details of these four different components of the reanalysis processing chain are provided in the subsections below, as well as the approach followed to monitor the assimilation of $CO_2$ and $CH_4$ satellite data.

### 110  2.2 Satellite GHG observations

The satellite measurements of radiances (L1 data) are processed by satellite retrievals developed by various data providers to derive information on the total and partial atmospheric column of $CO_2$ and $CH_4$ dry mole fraction (L2 data). In the CAMS GHG reanalysis only L2 products were used for $CO_2$ and $CH_4$. With nadir looking satellite instrument geometries the L2 data provide vertically integrated content with vertical sensitivity functions called either averaging kernel when an optimal
estimation approach (Rodgers, 2000) is used or weighting functions, that provide information on where the retrieval sensitivity is located along the vertical. The satellite products assimilated in this reanalysis are all provided with averaging kernel and prior information or weighting functions (Massart et al, 2014, 2016). Table 1 provides the specification for each of the assimilated satellite $CO_2$ and $CH_4$ products, selected as the state-of-the art retrievals at the beginning of 2017, when the CAMS GHG reanalysis production started. All of the L2 satellite products are freely available from the Copernicus Climate Change
Service (C3S) Copernicus Climate Data Store (Alos et al., 2019) at https://cds.climate.copernicus.eu/cdsapp#!/dataset/satellite-



carbon-dioxide for CO$_2$ and https://cds.climate.copernicus.eu/cdsapp#!/dataset/satellite-methane for CH$_4$. The GHG reanalysis integrate the L2 GHG data from the following satellite instruments:

- **SCIAMACHY – Envisat:** The The SCanning Imaging Absorption spectroMeter for Atmospheric CartograpHY (SCIAMACHY) instrument onboard the Envisat satellite was launched by the European Space Agency (ESA) in March 2002 and it was developed by a consortium involving the Netherland Space Office, the German Aerospace Centre and the Belgian Federal Science Policy Office. It measures radiances variations from the ultraviolet to the near visible infrared. The GHG L2 products use the nadir spectra of reflected and scattered solar radiation in the near-infrared region. Satellite radiance observations in the near infrared spectral region with the nadir looking geometry are sensitive to changes in CO$_2$ and CH$_4$ down to the Earth's surface. The measurements provide total column information with sensitivity peaking near the surface. The ground pixel size is typically between 30 km and 60 km and the swath width is about 960 km. There are no across-track gaps between the ground pixels but there are gaps along-track as SCIAMACHY operates only part of the time (approx. 50%) in nadir observation mode. The CO$_2$ and CH$_4$ column products are retrieved by the University of Bremen (Reuter et al., 2011) and the Netherland Institute for Space Research (SRON) (Frankenberg et al., 2011), respectively. Both of L2 products are delivered by the ESA GHG-Climate Change Initiative (Buchwitz et al, 2015) and the C3S Climate Data Store (https://cds.climate.copernicus.eu).

- **TANSO-FTS – GOSAT:** The Thermal And Near infrared Sensor for carbon Observations - Fourier Transform Spectrometer (TANSO-FTS) instrument onboard the Greenhouse Gases Observing Satellite (GOSAT) satellite has been developed by the Japan Aerospace Exploration Agency (JAXA) and it was launched in January 2009. TANSO-FTS measures radiances in the short-wave infrared band that provide information of total-column CO$_2$ and CH$_4$ mole fractions. Similar to SCIAMACHY, the sensitivity of the total column information provided by L2 data is peaking near the surface due to the spectral band used. The ground pixel size is about 10 km, the swath is 750 km and it has a revisit time of 3 days. In contrast to SCIAMACHY, the GOSAT scan pattern consists of non-consecutive individual ground pixels, i.e., the scan pattern is not gap-free. For a general overview about GOSAT see also http://www.gosat.nies.go.jp/en/. The L2 retrieval product is engineered by the SRON (Schepers et al., 2012, 2016) and delivered by the ESA GHG-CCI and the C3S Climate Data Store (https://cds.climate.copernicus.eu).

- **IASI – Metop A and B:** The Infrared Atmospheric Sounding Interferometer (IASI) instruments are onboard the Meteorological Operational satellites (Metop-A and Metop-B) launched in October 2006 and September 2012 respectively. The French National Centre for Space Studies (CNES) lead the design and developments of the instruments in collaboration with the European Organisation for the Exploitation of Meteorological Satellites (EUMETSAT). The IASI instruments measure the thermal infrared band with high spectral resolution enabling it to detect a wide range of trace gas variations in the atmosphere, including CO$_2$ and CH$_4$ sensitive in the mid and upper tropospheric regions between 5 and 12 km of altitude. IASI is an across track scanning system with a swath width of 2200 km, providing global coverage twice a day. The field of view is sampled by 2×2 pixels whose ground resolution is 12 km at nadir. Both CO$_2$ and CH$_4$ are engineered and delivered by the Centre National de Recherche Scientifique



(CNRS)-Laboratoire de Météorologie Dynamique (LMD) (Crevoisier et al., 2009a, 2009b, 2014). The two L2
       products are delivered by the ESA GHG-Climate Change Initiative (Buchwitz et al, 2015) and the C3S Climate Data
       Store (https://cds.climate.copernicus.eu).

**Table 1. Specifications of the satellite data used in the CAMS GHG reanalysis**

| Gas | Instrument - Satellite | Period assimilated | Version (data provider) | Reference | Peaking sensitivity |
|---|---|---|---|---|---|
| $CO_2$ | SCIAMACHY – Envisat | 20030101 – 20120324 | CO2_SCI_BESD (v02.01.02, IUP-UB) | Reuter et al., 2011) | Near Surface |
| | IASI – Metop-A | 20070701 - 20150531 | CO2_IAS_NLIS (v8.0, CNRS-LMD) | Crevoisier et al. (2009a) | Middle and Upper troposphere |
| | IASI – Metop-B | 20130201 – 20181130 | CO2_IAS_NLIS (v4.2_nrt, CNRS-LMD) | | Middle and upper troposphere |
| | | 20181201-20201231 | CO2_IAS_NLIS (v4.0_nrt, CNRS-LMD) | | |
| | TANSO-FTS - GOSAT | 20090601-20131231 | CO2_GOS_SRFP (V2.3.6, SRON) | Butz et al., (2011); Guerlet et al. (2013) ; Heymann et al. (2015) | Near Surface |
| | | 20140101-20181231 | CO2_GOS_SRFP(V2.3.8, SRON) | | |
| | | 20190101-20201231 | CO2_GOS_BESD (CAMS_NRT, IUP-UB) | | |
| $CH_4$ | SCIAMACHY – Envisat | 20030108-20100601 | CH4_SCI_IMAP (v7.2, SRON) | Frankenberg et al., (2011) | Near Surface |
| | IASI – Metop-A | 20070701-20150630 | CH4_IAS_NLIS (V8.3, CNRS-LMD) | Crevoisier et al., (2009b, 2014) | Middle and Upper troposphere |
| | IASI – Metop-B | 20130201- 20181130 | CH4_IAS_NLIS (V8.1_nrt, CNRS-LDM) | | Middle and upper troposphere |
| | | 20181201-20201231 | CH4_IAS_NLIS (v4.0_nrt, CNRS-LDM) | | |
| | TANSO-FTS - GOSAT | 20090601-20131231 | CH4_GOS_SRFP (V2.3.6, SRON) | Butz et al., (2010); Schepers et al., 2012 | Near Surface |
| | | 20140101-20181231 | CH4_GOS_SRFP (V2.3.8, SRON) | | |
| | | 20190101-20201231 | CH4_GOS_SRPR (CAMS_NRT, SRON) | | |

## 2.3 Surface fluxes and prescribed sources/sinks

The emissions and surface fluxes provide the surface boundary conditions for the atmospheric concentrations of $CO_2$ and $CH_4$.
They play a crucial role in determining the variability and growth rate of both greenhouse gases in the atmosphere. Errors in
the budget of the total flux will result into systematic errors or biases in forecast of atmospheric $CO_2$ and $CH_4$. In the CAMS
       reanalysis the surface fluxes (including sources and sinks) are not optimized by the assimilation system. This lack of surface



flux optimization can lead to biases in the analysis when the observing system coverage is sparse in space and time or when the observation error is large, and the analysis is strongly influenced by the model forecast.

Table 2 lists the datasets used to produce the CAMS reanalysis. They include:

- Fire emissions derived using the CAMS Global Fire Assimilation System (GFAS) version 1.2 that assimilate fire radiative power observations from satellite-based sensors (Kaiser et al., 2012). GFAS produces daily estimates of wildfire and biomass burning emissions. The emissions are injected at the surface and distributed over the boundary layer by the model's convection and vertical diffusion scheme.

- Anthropogenic emissions from the Emission Database for Global Atmospheric Research (EDGAR) version 4.2FT2010 inventory (Janssens-Maenhout et al., 2011; Olivier and Janssens-Maenhout, 2012) excluding the short carbon cycle. The anthropogenic emissions are based on annual average values and include fossil fuel combustion and leakage, agricultural, landfill/waste emissions and aviation (based on the Atmospheric Chemistry and Climate Model Intercomparison Project (ACCMIP, Lamarque et al., 2013) nitric oxide (NO) emissions from aviation scaled
to the annual $CO_2$ total emission from aviation from EDGAR). Anthropogenic emissions of $CO_2$ are extrapolated from 2010 to 2014 with the time series of country totals from EDGARv4.3 (Janssens-Maenhout et al., 2016) and from 2015 to 2020, a persistent growth based on the last available year (2014) is applied. $CH_4$ anthropogenic emissions are fixed with the last year of available gridded data (2010) from 2011 to 2020. Note that $CO_2$ and $CH_4$ emissions are not adjusted for the COVID emission reduction in 2020 (Le Quéré et al., 2020).

- Biogenic $CO_2$ fluxes are based on the online CHTESSEL module (Boussetta et al., 2013) that relates $CO_2$ biogenic fluxes with radiation, precipitation, temperature, humidity, and soil moisture. CHTESSEL is used in conjunction with the biogenic flux adjustment system (BFAS) that improves the continental budget of $CO_2$ fluxes by combining information from fluxes estimates by a global flux inversion system (Chevallier et al., 2010), land-use information and the CHTESSEL online fluxes (Agustí-Panareda et al., 2016).

- Wetland $CH_4$ monthly mean emissions come from a climatology (1990–2008) based on the LPJ-WHyMe model that is constrained by SCIAMACHY observations (Spanhi et al., 2011).

- A monthly modulation for $CH_4$ rice emissions is implemented based on the seasonal cycle of Matthews et al. (1991).

- The $CH_4$ chemical sink is represented by a monthly mean climatological loss rate from Bergamaschi et al. (2009)
based on OH fields optimized with methyl chloroform (Bergamaschi et al., 2005; Houweling et al., 1998) and stratospheric radicals from the 2D photochemical Max-Planck-Institute (MPI) model (Brühl and Crutzen, 1993).

- Other sources and sinks include a $CH_4$ monthly soil sink (Ridgwell et al., 1991), $CO_2$ and $CH_4$ annual mean oceanic fluxes (Houweling et al., 1999; Lambert and Schmidt, 1993; Takahashi et al., 2009) and $CH_4$ monthly mean fluxes from termites (Sanderson, 1996) and wild animals (Houweling et al., 1999).




**Table 2. Specifications of the emission and surface fluxes used in the CAMS GHG reanalysis**

| Gas | Emission/Flux type | Data provider - Version |
|---|---|---|
| $CO_2$ | $CO_2$ and $CH_4$ fire emissions | GFAS Version 1.2 (Kaiser et al., 2012) |
| | $CO_2$ ocean fluxes | Takahashi Climatology (Takahashi et al., 2009) |
| | $CO_2$ emissions from aviation | Based on ACCMIP NO emissions from aviation scaled to annual total CO2 from EDGAR aviation emissions (Olivier and Janssens-Maenhout, 2012) |
| | $CO_2$ ecosystem fluxes bias corrected with BFAS | Based on CHTESSEL (modelled online in IFS) (Boussetta et al., 2013; Agustí-Panareda et al., 2016) |
| | $CO_2$ anthropogenic emissions | EDGARv4.2FT2010 (2003-2010) (Olivier and Janssens-Maenhout, 2012) |
| $CH_4$ | $CH_4$ total natural emissions | based on EDGARv4.2FT2010 (2003-2010) (Olivier and Janssens-Maenhout, 2012); LPJ-HYMN wetland climatology  (Spahni et al., 2011);and other natural sources/sinks (Matthews et al., 1991; Ridgwell et al., 1999; Houweling et al., 1999; Lambert and Schmidt, 1993; Sanderson, 1996). |
| | $CH_4$ chemical sink | Monthly mean climatology of CH4 loss rate from Bergamaschi et al. (2009) |
| | $CH_4$ anthropogenic emissions | EDGARv4.2FT2010 (2003-2010) (Olivier and Janssens-Maenhout, 2012) |

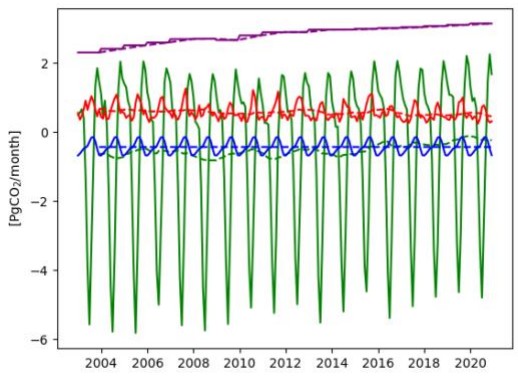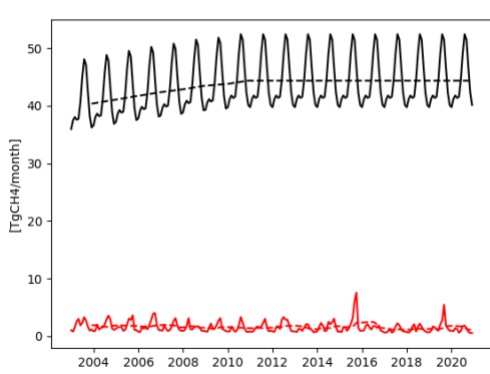


**Figure 3. Monthly $CO_2$ and $CH_4$ surface fluxes. The $CO_2$ fluxes [PgCO$_2$/month] include modelled Net Ecosystem Exchange (NEE) fluxes (in green), anthropogenic emissions (in purple), ocean fluxes (in blue) and biomass burning emissions (in red). The total $CH_4$ fluxes [TgCH$_4$/month] excluding biomass burning emissions are shown by black line and $CH_4$ biomass burning emissions [TgCH$_4$/month] are depicted in red. The dash lines show the 1-year running mean for each of the fluxes.**




## 2.4 Forecast model

The CAMS GHG reanalysis has been produced using the IFS model. The same model is used to produce operational numerical weather predictions (NWP) at ECMWF and the CAMS global forecast and analyses for reactive gas, aerosols and greenhouse gases at ECMWF (Fleming et al. 2015, Agustí-Panareda et al., 2017, Agustí-Panareda et al., 2022). The IFS model version used is IFS CY42R1, the same as in the CAMS reanalysis for reactive gases and aerosols (Inness et al., 2019). The forecasting model uses a reduced Gaussian grid with a resolution of Tl255 corresponding to a horizontal resolution of approximately 80 km and 60 hybrid-sigma pressure vertical levels from the surface to 0.1hPa. The tracer advection is computed using a semi-implicit semi-Lagrangian scheme (Temperton et al., 2001; Diamantakis and Magnusson, 2016) that is not mass-conserving. This scheme leads to an error growth that can dominate the signal in the model simulations if it is not corrected. Thus, a mass fixer is required to ensure mass conservation at every time step (Diamantakis and Agustí-Panareda, 2017). The mass fixer is particularly important for long-lived greenhouse gases for which the interesting signals to monitor, e.g., trends or annual growth rates and large-scale spatial gradients, are weak compared to the large background values. The transport model also includes a turbulent mixing scheme (Sandu et al., 2013) and a convection scheme (Bechtold et al., 2014). For the $CH_4$ chemical sink in the troposphere and the stratosphere, climatological loss rates derived from the Max Planck Institute photochemical model are used (Bergamaschi et al., 2009). Full documentation of the IFS can be found at https://www.ecmwf.int/en/forecasts/documentation-and-support/changes-ecmwf-model/ifs-documentation.

## 2.5 Analysis procedure (data assimilation)

The IFS system is using an incremental formulation of the 4-dimensional variational technique (4D-Var). The 4D-Var technique consists of minimizing a cost function that combines the model information and the observation information in order to obtain the best possible state of the atmosphere (analysis) accounting for the model and observation errors. The incremental 4D-Var cost function is quadratic and is formulated as follows:

$$J(\delta x) = \frac{1}{2}(\delta x - \delta x_b)^T B^{-1}(\delta x - \delta x_b) + \frac{1}{2}(G\delta x - d)R^{-1}(G\delta x - d) \tag{1}$$

where $\delta x$ is the increment i.e., the difference between the model state $x$ and the first guess $x_g$, $\delta x_b$ is the difference between the background (the forecast started from the previous analysis) and the first guess, $B$ the background error covariance matrix, $R$ the observation error covariance matrix, $G$ the observation operator or forward operator that translate the information from model space to observation space. The innovation vector is $d = y - Gx_g$ with $y$ the observation vector and $x_g$ the first guess. When the minimization of the cost function is complete, $\delta x$ is added to $x_g$ to provide the analysis.





$$x_a = x_g + \delta x \qquad\qquad (2)$$

245  The analysis is performed over 12-hour assimilation windows from 9:00 to 21:00 and from 21:00 to 9:00 UTC. The incremental
4D-Var assimilation involves the stepwise minimization of the linearized cost function (equation 1) by updating the first guess
$x_g$ and increasing the resolution. In the CAMS reanalysis setup, two minimizations are completed successively at TL95
(approximately 210 km) and TL159 (approximately 110 km) spectral truncations. Once the assimilation procedure is
completed an analysis is generated that will serve to initialize the next forecast at the full TL255 resolution.

250

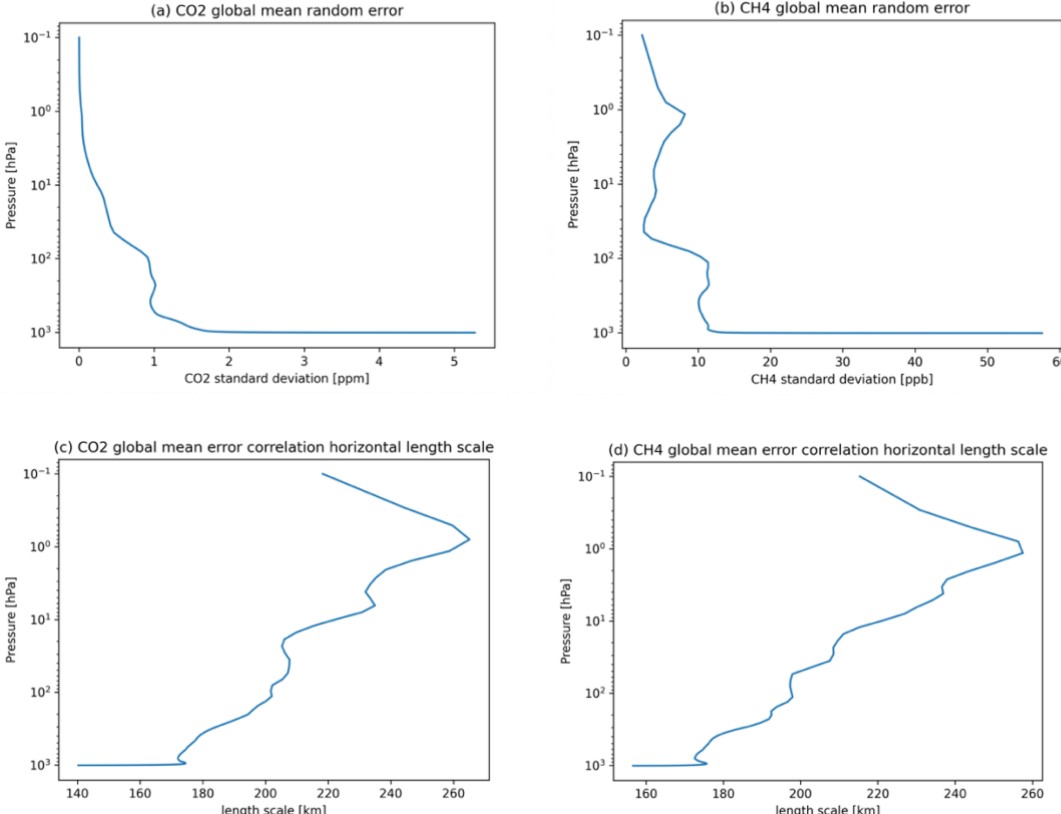

**Figure 4.** Model background error for $CO_2$ and $CH_4$ used in the CAMS GHG reanalysis: (a,b) global mean standard deviation and (c,d)
global mean error correlation length scale across the vertical levels.

255

The background errors for $CO_2$ and $CH_4$ were produced from an ensemble of data assimilations (Massart et al., 2016), which
allows the calculation of differences between pairs of background fields which have the characteristics of the background
errors. The background errors for the greenhouse gas species are univariate, which means that there is no correlation between
the greenhouse gas species and the dynamical fields. Hence each species is assimilated independently from the others. The





background errors used for both the greenhouse gas species and the dynamical fields are also constant in time. In the ECMWF data assimilation system, the background error covariance matrix is given in a wavelet formulation (Fisher, 2004, 2006). This allows both spatial and spectral variations of the horizontal and vertical background error covariances globally. Figure 4 shows the global mean of the standard deviation and average horizontal correlation length scales for both $CH_4$ and $CO_2$. Following experimentation, the correlation length scales between the background errors were manually reduced in the atmospheric boundary layer (1km from the surface).

## 2.6 Monitoring the data assimilation system

The time series of the departures (or differences) between the analysis (AN) and the assimilated satellite data (hereafter referred to as observations, OBS), as well as those between the underlying model simulation (or background, BG) and the observations, are used to monitor the performance of the analysis system and are shown in Figures 5 (for $CO_2$) and 6 (for $CH_4$). For each satellite retrieval product, both the BG departures (OBS-BG, green lines) and the AN departures (OBS-AN, red lines) are plotted (panel a: SCIAMACHY, panel b: IASI-A; panel c: IASI-B; panel d: GOSAT), together with the number of observations assimilated monthly (blue lines). Overall, both the random (i.e., standard deviation, dashed lines) and the systematic components of the departures (i.e., average values, solid lines) are shown to be reduced by the assimilation process, as highlighted by the AN departures (red lines) being closer to zero than the BG departures (green lines). Note that the difference between the BG and the AN departure is equal to the analysis increments associated with the related observations (i.e. AN-BG).

The number of observations assimilated is different for each satellite instrument and varies with time: IASI generates the largest number of data, with both instruments (IASI-A and IASI-B) providing between 150 000 and 200 000 $XCO_2$ or $XCH_4$ data per month; the observations taken by SCIAMACHY oscillate between 25 000 and 50 000 for $CH_4$ and between 5 000 and 10 000 for $CO_2$; the number of GOSAT $XCO_2$ data fluctuate around 2 500, whereas those from GOSAT $XCH_4$ are comprised between 5 000 and 10 000 per month. It is also clear from Figs 5(a,d) and 6(a,d) that fewer $XCO_2$ and $XCH_4$ data from SCIAMACHY, IASI and GOSAT-TANSO are assimilated during the winter months. A magenta vertical dashed line in Figs 5(c,d) and 6(c,d) indicates when the near-real time satellite products started to be assimilated in early 2019. This transition produced an abrupt change in the quality and availability of both IASI and GOSAT retrievals.

The modelled $XCO_2$ is systematically larger than the observations (leading to overall negative BG departures) because of the biases in the total fluxes (see section 2.3). Therefore, all instruments produced negative departures until 2013 (Fig. 5). From 2013 to 2019, the modelled values of $XCO_2$ became smaller than those measured by GOSAT (Fig. 5(d)), while the model continued to (slightly) overestimate the IASI $XCO_2$ observations in the mid to upper troposphere. This overestimation is



consistent with a drift in the IASI $CO_2$ data towards a growing negative bias. After 2018, part of the drift is due to the fact that IASI (version v4.0) is saturating with increasing atmospheric $CO_2$. Note that this has been corrected with v9.1 (available on

the C3S datastore). A sudden change in the IASI-B $XCO_2$ departures is visible in Fig. 5(c) around December 2018, in correspondence of the switch from the ESA-CCI reprocessed dataset to a near-real time LMD dataset used operationally in the CAMS GHG analysis. The transition to a new dataset was made necessary as the reanalysis production was running close to real-time and reprocessed observations were not available. After the transition to near-real time observations, the IASI $XCO_2$ increments are reduced to almost zero, as hinted by the overlap between the red (AN departure) and green line (BG departure)

in Fig. 5(c). At the same time, a drop in the number of assimilated IASI $XCO_2$ observations is observed (blue line, same panel and figure). Together with a drastic reduction in the magnitude of the increments, a large negative bias of approximately 5ppm in both the AN and BG departures emerges, accompanied by a large standard deviation error (~20 ppm, cf. dashed lines in same panel and figure). This degradation in the quality of the IASI-B $XCO_2$ observations in the near-real time dataset is due to the change of the correction of the non-linearity of the detector of IASI-B that was made by CNES and EUMETSAT on

August 17$^{th}$ 2018 and that introduced a bias of ~0.2 K on the channels used to perform the $CO_2$ retrieval. This change has been corrected in the versions of IASI-B MT-CO2 that are available on the C3S datastore but were not used for this reanalysis. In January 2019, there was also a transition from the ESA-CCI GOSAT $XCO_2$ retrievals to the near-real time IUP-UB retrieval product (Heymann et al., 2015; Massart et al. 2016) as shown in Fig. 5(d). Consequently, the standard deviation of both the AN and the BG departures increases (cf. dashed lines, same panel and figure), suggesting that the near-real time data is noisier

than the reprocessed dataset from ESA-CCI.

The mean $XCH_4$ departures (both AN and BG) of SCIAMACHY and IASI are relatively small (a few ppb) compared to GOSAT (up to 10 ppb), throughout the entire time period (see solid red and green lines in Fig. 6). The XCH4 SCIAMACHY data was not used from 9 April 2012 onwards (Fig. 6(a)). The standard deviation of both the AN and BG departures are smaller

for GOSAT (around 10 ppb, dashed lines in Fig. 6(d)) than for SCIAMACHY (around 20 ppb, dashed lines in Fig.6(a)), indicating that GOSAT provides less noisy observations. Similar to what was observed for $CO_2$, a discontinuity in the mean AN and BG departures of GOSAT $XCH_4$ emerges in January 2019, in correspondence of the transition from the ESA-CCI dataset and the NRT SRON retrievals (see dashed pink line in Fig. 6(d)). Both the AN and the BG departures change sign, indicating that while up to 2019 both the analysis and model were underestimating the GOSAT observations, they start to

overestimate them since 2019. Since there was no modification to the model used for the reanalysis over this period, the cause of this negative bias emerging in both the AN and the BG departures since 2019 can only be attributed to the NRT GOSAT $XCH_4$ observations, and in particular to the fact that they are generated by using an extrapolated $XCO_2$ value in the proxy retrieval. In addition, the number of assimilated NRT GOSAT $XCH_4$ observations approximately doubles (blue line in Fig. 6(d)). Note that the switch to the near-real time retrievals for IASI-B XCH4 has a much more marginal impact on the system

(Fig. 6(c)).





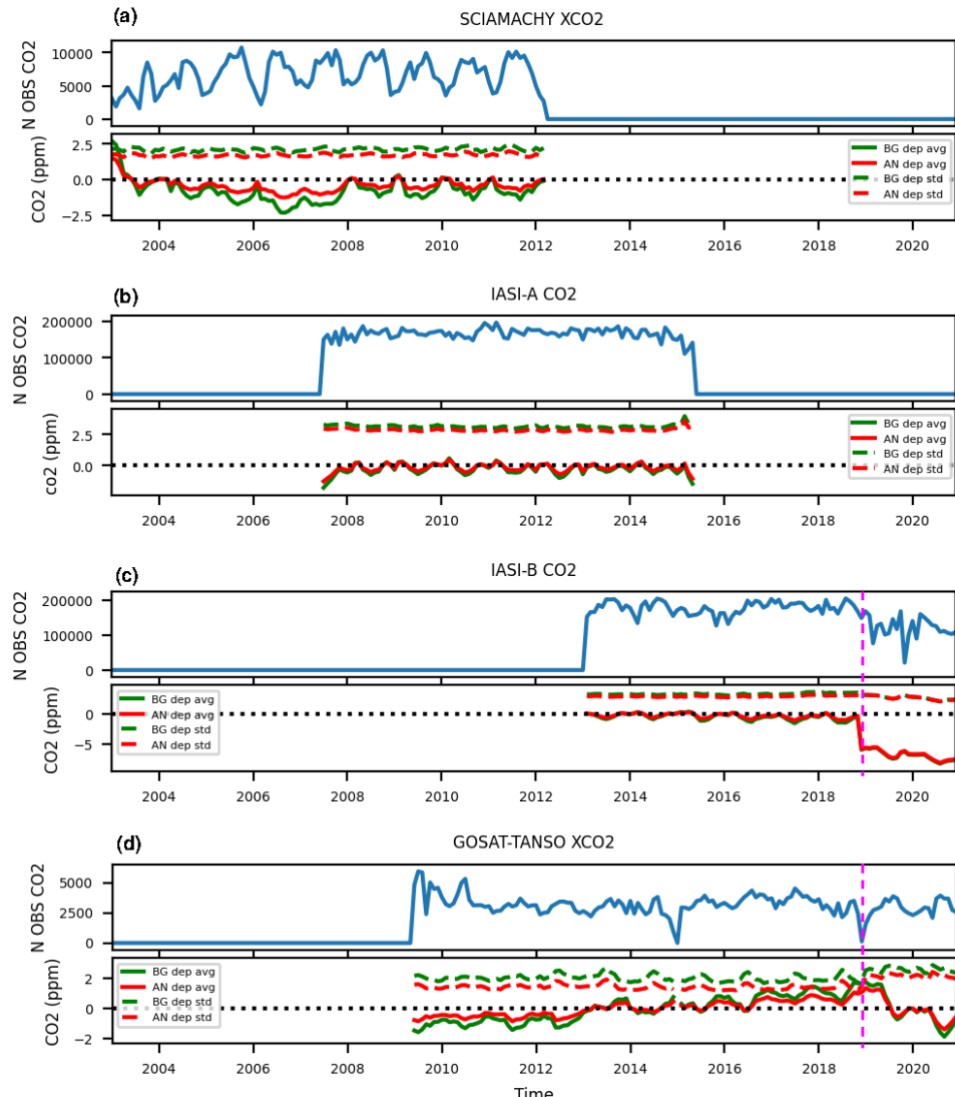

**Figure 5. Time series of global monthly number of XCO₂ satellite data (blue) and monthly mean CO₂ analysis (AN) and model background (BG) departures of the various observations (OBS) assimilated in the reanalysis (AN-OBS and BG-OBS in red and green respectively, see legend). The solid lines show the monthly average of the departures, and the dash lines the monthly standard deviations. The magenta dash line indicates the switch to the near-real time satellite products. Note that the range of values in y-axis varies depending on the satellite product.**


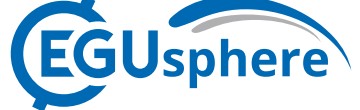

**Figure 6. Time series of global monthly number of XCH₄ satellite data (blue) and monthly mean CH₄ analysis (AN) and model background (BG) departures of the various observations (OBS) assimilated in the reanalysis (AN-OBS and BG-OBS in red and green respectively, see legend) for different satellite products. The solid lines show the monthly average of the departures, and the dash lines the monthly standard deviations. The magenta dash line indicates the switch to the near-real time satellite products. Note that the range of values in y-axis varies depending on the satellite product.**






## 3 Evaluation with independent observations

Validation against a set of independent observations has been performed on the 18 years of the CAMS GHG reanalysis span.
The independent data includes different types of observations: in situ near-surface continuous observations of $CO_2$ and $CH_4$
mole fractions from the collaborative ObsPack datasets (Schuldt et al., 2020; Sarra et al., 2021; NOAA Carbon Cycle Group
ObsPack Team, 2019; see Table A1); dry-air column-averaged mole fractions from the Total Carbon Observing Network
(TCCON, Wunch et al. 2011, 2015); tropospheric and stratospheric partial columns for $CH_4$ from the Network for the Detection

of Atmospheric Composition Change (NDACC, De Mazière et al., 2018) (see Table A2); AirCore vertical profiles of $CO_2$ and
$CH_4$ mole fractions (Karion et al., 2010; Baier et al., 2021); and the NOAA global mean $CO_2$ and $CH_4$ mole fraction product
based on the Greenhouse Gas Marine Boundary layer Reference (Conway et al., 1994, Dlugokencky et al., 1994, Massarie et
al., 1995).

### 3.1 Surface and column data

### 3.1.1 Carbon dioxide

Overall, the error is within ±10 ppm and ±4 ppm for most of the near-surface and total column stations respectively for the
whole 18-year period (Figs 7 and 8). Near the surface (Fig 7), there is a large variability in the $CO_2$ error between continental
stations influenced by local fluxes (e.g., CDL, FSD, AMT, HUN, see Table A.1) and oceanic stations sampling well-mixed air

(ALT, BRW, MHD). Continental stations show large error variations with season (e.g., CDL, HUN), with an underestimation
of $CO_2$ in the summer and an overestimation in the winter, indicating an underestimation of the amplitude of the $CO_2$ seasonal
cycle largely driven by vegetation growth. Differences between stations will be determined by the footprint of observations
having different contributions of fluxes from different biomes and from anthropogenic emissions. Accuracy of such fluxes can
vary geographically.


Overall, there is positive bias of a few ppm between 2003 and 2015 in the baseline surface stations (e.g. BRW, SMO, SPO)
which is consistent with the $XCO_2$ error at the TCCON sites (Fig. 8). This positive bias decreases from 2007 to 2015 when
IASI-A $CO_2$ data are assimilated, with values lower than 2 ppm, and becomes negative from 2015 to 2019 (from 0 to -2 ppm).
From 2019 onwards, there is a positive trend in the bias, and it becomes positive (> +2 ppm) in 2020. There is consistency

between the column and surface biases with a general positive bias at background stations before 2015 and a negative bias
after 2015 (up to 2019) at the surface stations, although there is no data in 2020 from the surface stations.



The synoptic and large-scale variability of $CO_2$ is well represented by the reanalysis (lower panel in Fig 8). The root mean square error at TCCON stations is below 0.8 ppm for $XCO_2$. The normalised standard deviation is around 1.0 (+/- 0.3) and the

Pearson correlation coefficient is larger than 0.8.

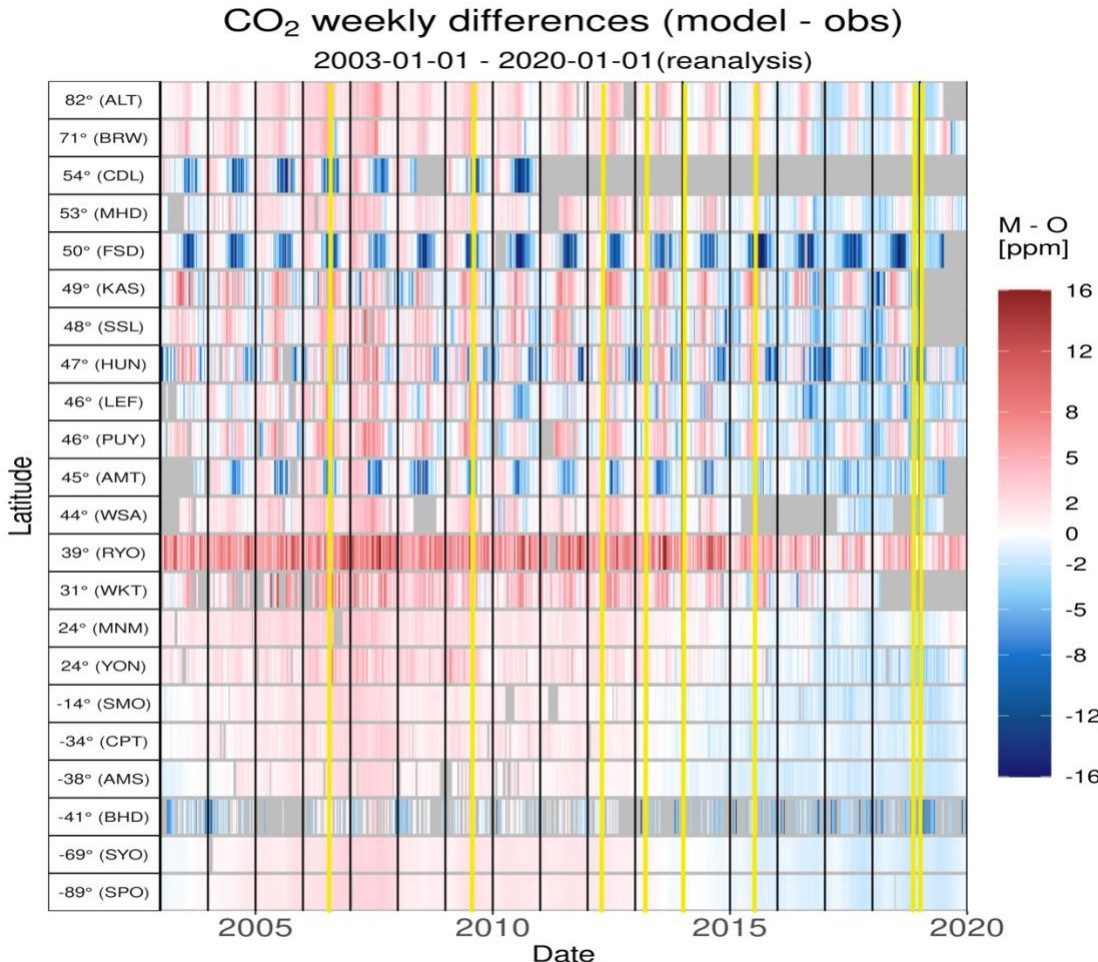

**Figure 7. Mosaic plot of $CO_2$ weekly biases (in ppm) of the CAMS GHG reanalysis compared to surface continuous observations of $CO_2$ mole fraction obtained from GLOBALVIEWplus $CO_2$ ObsPack v6.0 (Schuldt et al., 2020) and listed in Table A1. Each coloured**
**vertical line represents a weekly mean. Vertical yellow lines depict the changes in the assimilated data documented in Figs 1, 5 and 6. Grey shading indicates no observations are available.**



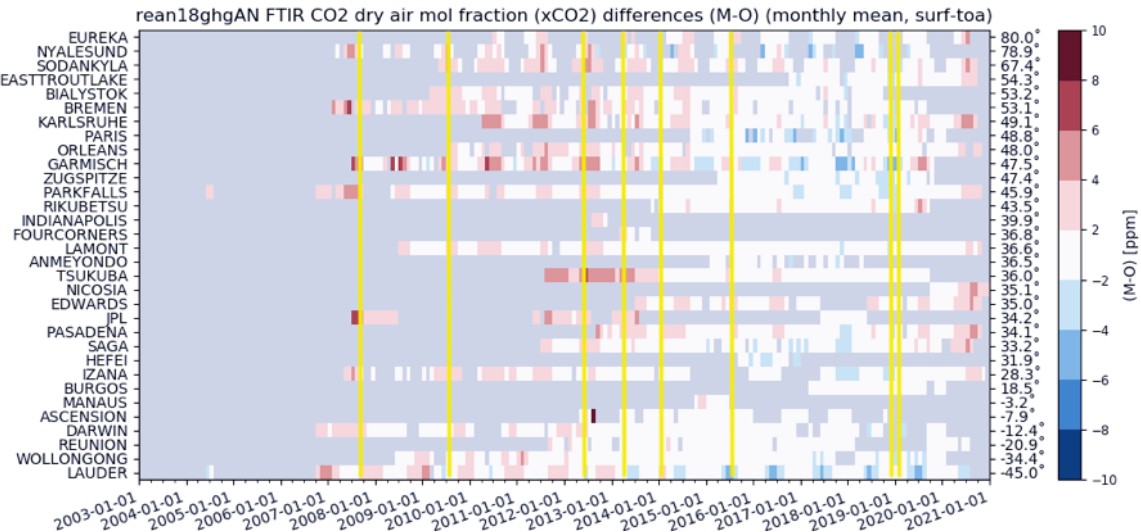

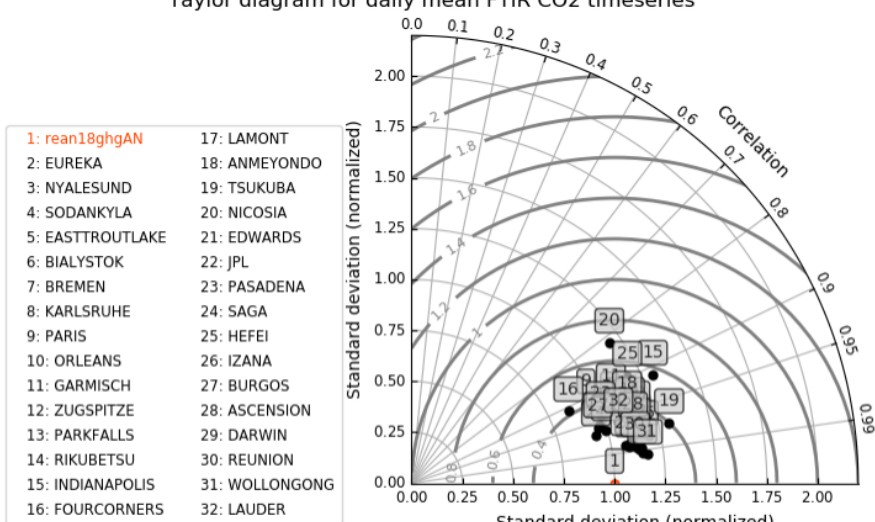

**Figure 8. Top: Mosaic plot of the CAMS GHG reanalysis biases at all TCCON sites (see Table A2) for the column-averaged dry mole fraction of $CO_2$ [ppm] (XCO2) averaged daily around local noon (+/- 2.5 hours). Vertical yellow lines depict the changes in the assimilated data documented in Figs 1, 5 and 6. Grey shading indicates no observations are available. Bottom: Taylor diagrams for the station dependent XCO₂ comparison of the CAMS GHG reanalysis against TCCON FTIR data.**







### 3.1.2 Methane

The $CH_4$ reanalysis fields are generally in good agreement with surface and tropospheric column observations with typical weekly and monthly errors within ±40 ppb and ±25 ppb respectively (Figs 9, 10 and 11). Stratospheric partial columns compared to NDACC data reveal a positive bias that is of the same order as the reported measurement uncertainty of 7% (Fig.

10, upper panel). The averaged relative differences in the troposphere across all NDACC sites are -0.4% for the reanalysis (Fig.10, lower panel), which is well within the measurement's uncertainty. The reanalysis overestimates the column-averaged $CH_4$ compared to TCCON observations (Fig. 11), for most mid- and high-latitude sites, with a relative difference of up to 2.5%, but shows a good agreement for the low latitude sites at Izaña, Darwin and Wollongong.  At the surface the bias is overall positive up to 2007 (Fig. 9). With the introduction of IASI, the biases are reduced. However, with the switch to near-

real time satellite data, the bias become negative at all sites reaching values lower than -20 ppb.

Differences between the surface and total column biases stem from the fact that the model suffers from large positive biases above the tropopause (between 100hPa and 10 hPa) of about 80-100 ppb during the months between September and November (Figs 5d and 6d of Verma et al., 2017) which affect the total column average. This stratospheric bias cannot be corrected systematically by $CH_4$ satellite data from SCIAMACHY, GOSAT and IASI.

For all observations (surface, partial and total columns) $CH_4$ shows a seasonality in the relative difference between observations and the reanalysis. The magnitude of the difference is site dependent. During local autumn/winter months the relative bias is increased (underestimation) at most surface sites and in the tropospheric columns. This underestimation is also seen in the TCCON time series. In the spring and summer there is an overestimation of $CH_4$ near the surface and in the total column. These biases are related to errors in the seasonal cycle of surface emissions, most likely from agriculture and wetlands, and

the accuracy of the representation of the hydroxyl radical (OH) sink which overall have larger values in the climatology compared to CAMS IFS(CB05BASCOE) atmospheric chemistry model OH (Segers et al., 2020b, Williams et al., 2021).  The $XCH_4$ root mean square error is around 1.4 ppb and the Pearson correlation coefficient is larger than 0.7 for $XCH_4$ except for some outliers (Fig. 11, lower panel), indicating a good representation of the synoptic variability, as for $XCO_2$.




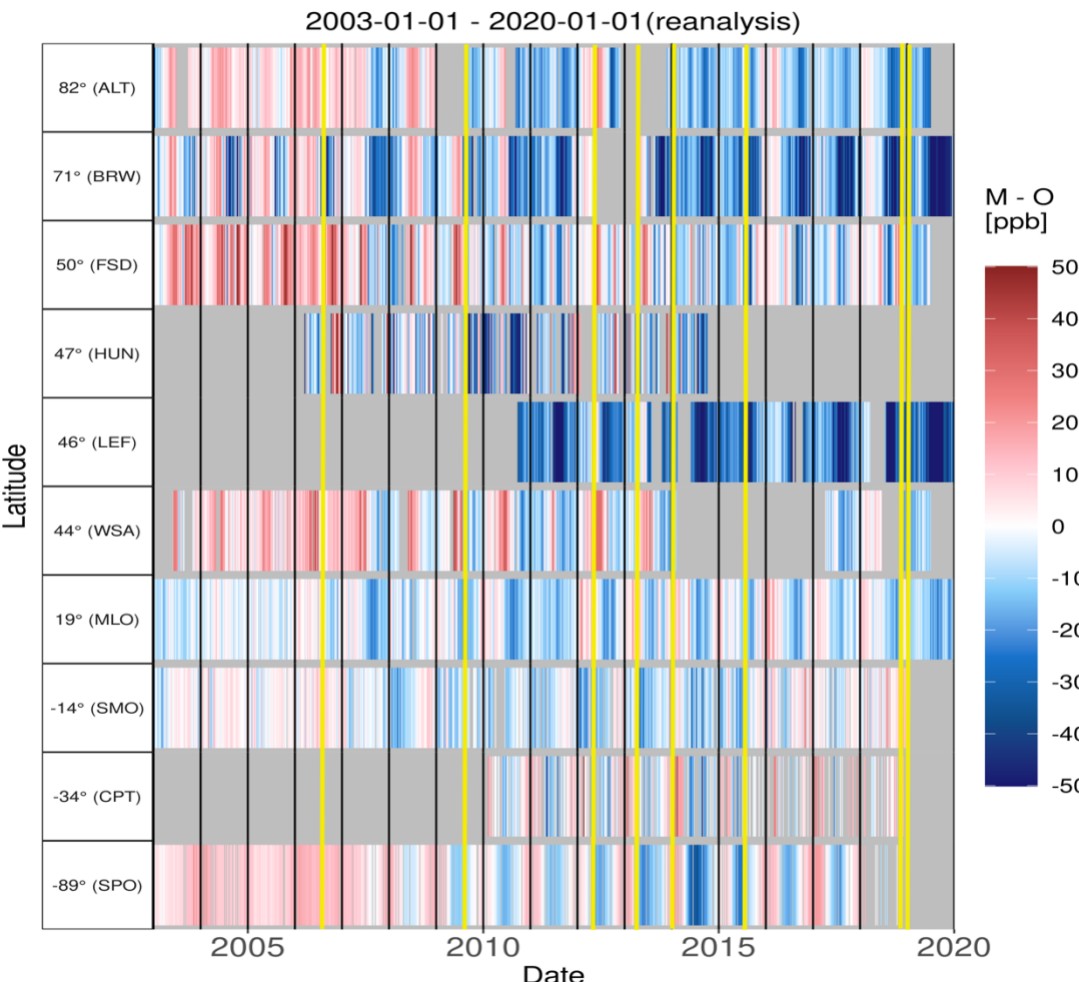

**Figure 9. Mosaic plot of CH₄ biases (in ppb) compared to surface continuous observations from GLOBALVIEWplus CH4 ObsPack v1.0 data product (Cooperative Global Atmospheric Data Integration Project, 2019) listed in Table A1. Each coloured vertical line represents a weekly mean. Vertical yellow lines depict the changes in the assimilated data documented in Figs 1, 5 and 6. Grey shading indicates no observations are available.**







**Figure 10. Mosaic plot of seasonal relative CH₄ biases at all FTIR sites (see Table A2) for the stratospheric columns (top) and tropospheric columns (bottom) NDACC. Vertical yellow lines depict the changes in the assimilated data documented in Figs 1, 5 and 6. Grey shading indicates no observations are available.**





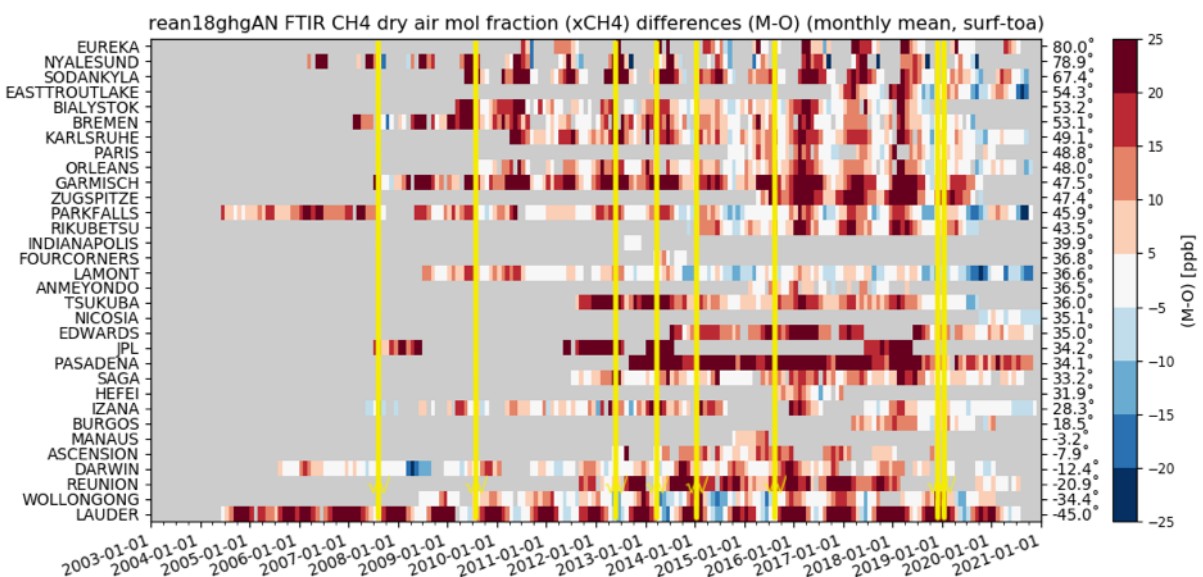

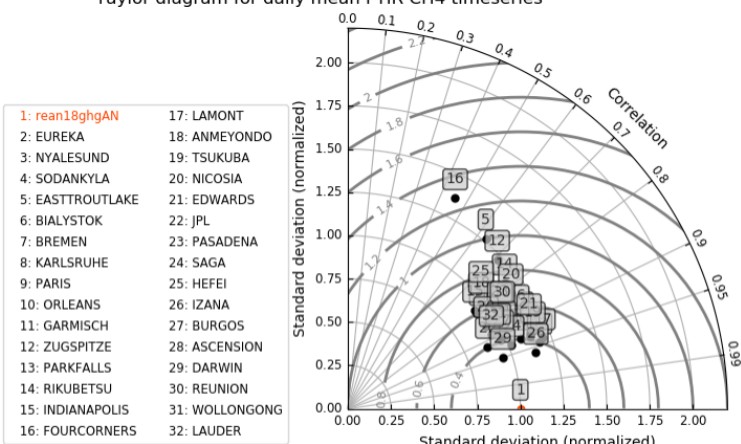

**Figure 11 Top: Mosaic plot of monthly biases at all TCCON sites for the column-averaged mole fractions XCH₄ [ppb] averaged**
**daily around local noon (+/- 2.5 hours). Vertical yellow lines depict the changes in the assimilated data documented in Figs 1, 5 and**
**6. Grey shading indicates no observations are available. Bottom: Taylor diagrams for the station dependent XCH₄ comparison of**
**the CAMS GHG reanalysis against TCCON FTIR data.**




## 3.2 Vertical profiles


The uncertainty of CAMS GHG reanalysis varies with height and the accuracy of the analysis vertical profiles depends mostly on the underlying model uncertainty, as the satellite data assimilated in reanalysis only provide integrated total or partial atmospheric column. The reanalysis has been evaluated using observations of $CO_2$ and $CH_4$ vertical profiles (Karion et al., 2010; Baier et al., 2021) from the NOAA AirCore dataset v20210813. It includes 133 vertical profiles from the surface to the

lower stratosphere (up to around 40 hPa) from 2012 to 2020 at 7 sites listed in Table A.3.

 Figure 12 shows that the largest mean error occurs (i) near the surface with a strong influence from surface fluxes; (ii) in Upper Troposphere/Lower Stratosphere (UTLS) region (between 500 hPa and 100 hPa) with a strong influence from long-range transport; and (iii) in the stratosphere (above 100 hPa) where uncertainties associated with chemical loss of $CH_4$ and the

meteorology driving the tracer transport are largest, and the fact that satellite data used here are not able to constrain the stratospheric $CO_2$ and $CH_4$ in the reanalysis. Near the surface, there is a positive $CO_2$ bias associated with an overestimation of the total flux in the model and a negative $CH_4$ bias which stems from both errors in the emissions and the chemical loss rate in the troposphere. The negative $CO_2$ bias in the UTLS agree with the tendency of the model to underestimate fine-scale higher-valued $CO_2$ streamers associated with long-range transport. The large positive $CH_4$ bias in the stratosphere of around 200 ppb

is consistent with the positive biases with respect to NDACC stratospheric column (Fig. 10, upper panel) and the documented model biases with respect to MIPAS and ACE-FTS by Verma et al. (2017). The errors associated with the stratospheric chemical sink are thought to be the largest contributor to the stratospheric $CH_4$ bias as shown by tests using the IFS BASCOE-CB05 chemical loss rate (not shown here). In general, the reanalysis underestimates the $CO_2$ vertical gradient across the tropopause. This underestimation leads to a positive bias for $CO_2$ in the lower stratosphere of around 2 ppm. The analysis is

not able to remove the large errors near the surface by only adjusting the atmospheric mole fractions, i.e., without adjusting the emissions in the data assimilation process, nor it is able to reduce the stratospheric errors in the model (Massart et al. 2017, Verma et al. 2017). The vertical profiles have a large variability from day to day as shown in Figure 13 with a sequence of profiles at Traînou (France). The CAMS GHG reanalysis is able to capture these synoptic variations in the vertical profile, consistent with its skill to represent $XCO_2$ and $XCH_4$ synoptic variability (lower panels of Figs 8 and 11).




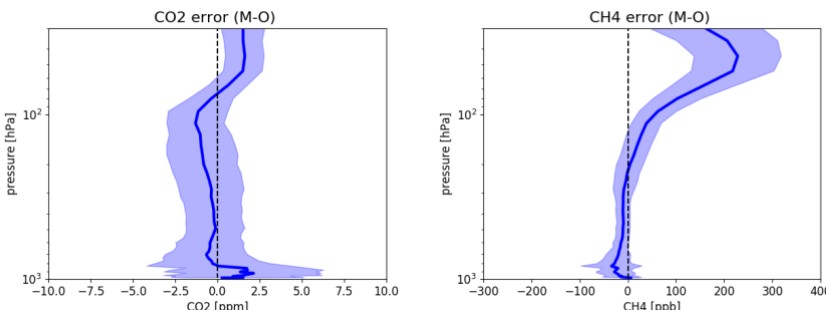

**Figure 12: Vertical profiles of mean error (Model M- Observation O) of CAMS CO₂ (left) and CH₄ (right) reanalysis with respect to AirCore observations comprising 133 profiles at 7 sites (listed in Table A3) over the period from 2012 and 2020. The blue shading shows the +/- standard deviation of M-O with respect to the mean error. The vertical dash black line depicts the zero mean error.**


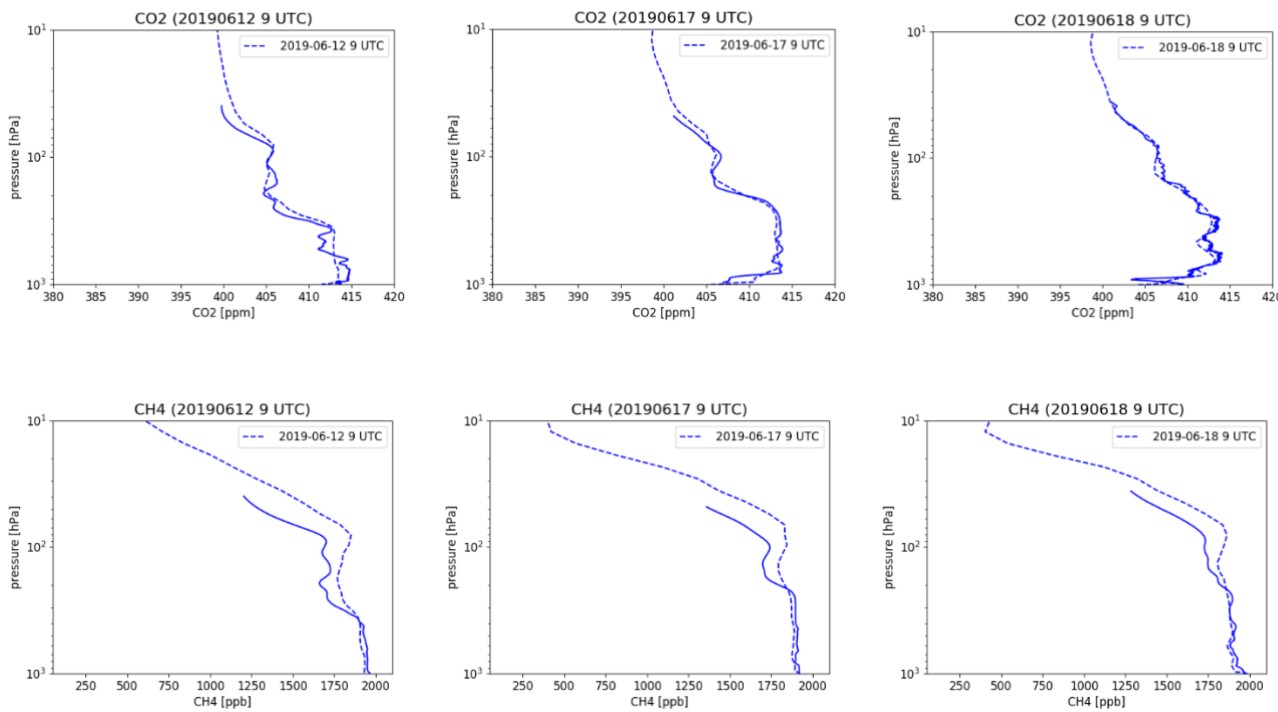


**Figure 13: Vertical mole fraction profiles of CO₂ and CH₄ from the CAMS GHG reanalysis (dash line) and AirCore observations (solid line) at Traînou (France, see Table A3) over the period in June 2019.**




### 3.3 Trends

Although this reanalysis is using a consistent underlying model and re-processed observations of $CO_2$ and $CH_4$, the current
system is not able to provide accurate enough atmospheric mole fraction that can be used to estimate trends and the atmospheric growth rate as computed by the changes in global mean $CO_2$ and $CH_4$ from one year to the next. The $CO_2$ and $CH_4$ global annual means based on Marine Boundary Layer (MBL) reference sites are compared to the NOAA Global Greenhouse Gas Reference Network (GGGRN) observations (https://gml.noaa.gov/ccgg/about.html, Andrews et al., 2014; Conway et al., 1994; Dlugokencky et al., 1994) in Fig. 14. Changes in the assimilated satellite data have a clear impact on the evolution of the global
annual mean values of $CO_2$ and $CH_4$ in the CAMS GHG reanalysis. The reanalysis has a positive global bias in near-surface $CO_2$ and $CH_4$ of a few ppm and around 20 ppb respectively from 2003 to 2007. Note that this positive bias in the annual global mean does not imply that the bias will be positive everywhere, as shown by the negative surface $CH_4$ biases at the AirCore sites (Fig. 12) and the large temporal and geographical variability of the weekly bias illustrated in Figs 7 and 9. After the introduction of IASI in 2007 the global bias decreases and it is lowest during the period when the number of observations is
largest in 2013 and 2014 (Figs 5 and 6). Finally, the change to the near-real time satellite retrievals in 2019 together with the incorrect trend in the emissions during the COVID slowdown period in 2020 (Le Quéré et al., 2020) lead to changes in the global bias from negative to positive for $CO_2$ and from positive to negative for $CH_4$. These changes in the global bias are consistent with the changes in the errors with respect to total-column and near-surface observations in Figs. 7 to 11. It is important to note that the changes in global bias associated with changes in the assimilated data are of the same order of
magnitude     as     the     observed     atmospheric     growth     rate     of     $CO_2$     (gml.noaa.gov/ccgg/trends)     and     $CH_4$ (gml.noaa.gov/ccgg/trends_ch4). For this reason, this reanalysis product is not suitable for trend analysis.

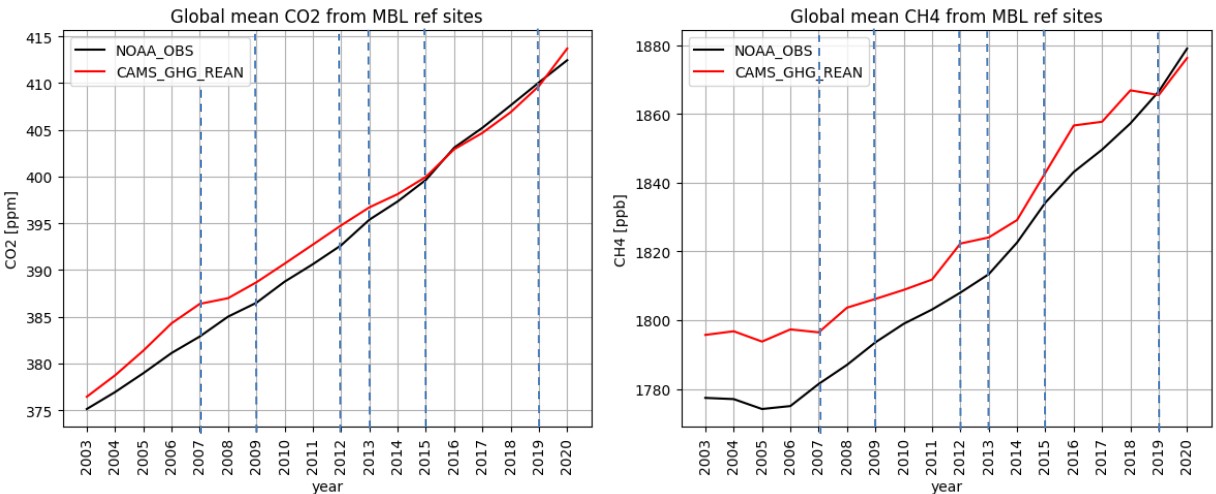

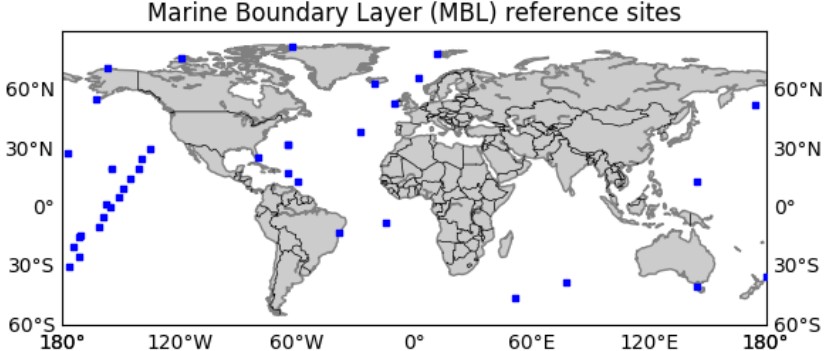

**Figure 14: Global mean CO₂ [ppm] and CH₄ [ppb] from the CAMS GHG reanalysis (in red) and the NOAA global mean CO₂ and CH₄ (in black, https://gml.noaa.gov/ccgg/trends/global.html) based on the Greenhouse Gas Marine Boundary layer Reference (Conway et al., 1994, Dlugokencky et al., 1994, Massarie et al., 1995, Dlugokencky et al., 2021). The global mean of the CAMS GHG reanalysis has been computed based on the same NOAA Marine Boundary Layer (MBL) reference sites shown in the map (bottom panel, see https://gml.noaa.gov/ccgg/mbl/mbl.html for further details). The dash blue lines mark the years when there was a change in the observing system. The uncertainty associated with the computation of global mean using the MBL sites is estimated to be 0.1 ppm for CO₂ (Ed Dlugokencky and Pieter Tans, NOAA/GML, gml.noaa.gov/ccgg/trends/) and below 2 ppb for CH₄ (Ed Dlugokencky, NOAA/GML (gml.noaa.gov/ccgg/trends_ch4/).**

## 4. Limitations and caveats

This section provides an overview of the shortcomings of the CAMS GHG reanalysis which users should consider when interpreting the data. The main issues documented in the previous sections are summarised below:

1. Emissions are prescribed and not adjusted by the data assimilation system in the CAMS reanalysis (Sect. 2.3). This leads to a growing model error for $CO_2$ and $CH_4$ that can be difficult to correct with a sparse observing system and 12-hour assimilation window. In addition, the prescribed emissions are not available in near-real time, which means they are either kept fixed since the last year available (e.g. 2010 for $CH_4$) or they are extrapolated with a climatological trend as done for $CO_2$ (see details in Sect. 2.3). Because of this, the CAMS GHG reanalysis is not suitable to investigate the impact of local emission changes, such as COVID impact studies, which require a large local emission adjustments to the prescribed inventories (e.g. Doumia et al., 2021) and atmospheric inversion systems to estimate the changes (e.g. McNorton et al, 2022).

2. Changes of satellite data used with different temporal, horizontal and vertical coverage cause changes in the quality of the reanalysis. For example, winter seasons have a lower number of observations because of light conditions and the higher frequency of cloud presence. This affects the quality of the seasonal cycle and the inter-hemispheric gradient. Similarly, in regions where there is no observation coverage, such as the stratosphere, the reanalysis is based on the underlying model including its systematic errors (see discussion on stratospheric biases in Sect 3.2).





3. Changes in satellite retrievals affect the quality of the observations used in the CAMS GHG reanalysis. For example, the switch from the CCI re-processed satellite products to the near-real time products is associated with a marked change in the bias and random error (i.e. standard deviation) of the departures from $XCO_2$ and $XCH_4$ GOSAT observations, as well as in the bias of the departures from the $XCO_2$ IASI-B observations. This large increase in the bias of the assimilated $CO_2$ and $CH_4$ observations from 2019 onwards results into a large increase in the bias of the CAMS GHG reanalysis in 2019 and 2020 which has implications for the trend analysis (Sect. 3.3).

4. The fixed climatological chemical loss rate of $CH_4$ (Sect 2.3) has been shown to overestimate the atmospheric $CH_4$ chemical sink by Segers et al. (2020b). Preliminary tests coupling the IFS to the atmospheric loss rate derived from BASCOE-CB05 chemistry have indeed shown a large reduction in the $CH_4$ negative bias in mid-latitudes. Systematic errors in the $CH_4$ chemical sink used in this reanalysis may have contributed further to enhance the large negative $CH_4$ bias in the CAMS GHG reanalysis over the last period in 2020, when the increase in the $CH_4$ growth rate has been linked to a decrease in chemical loss rate (Stevenson et al., 2021).

5. The large $CH_4$ and $CO_2$ biases in the stratosphere are currently under investigation. The $CH_4$ stratospheric bias is mainly associated with the use of a climatological loss rate (Sect 2.3), as preliminary tests using a different chemical loss rate based on IFS CB05BASCOE simulations show that the bias in $CH_4$ is greatly reduced.

6. Changes in systematic errors with time due to model error and changes in observation coverage and quality will affect trend analysis (see Sect. 3.3).

An up-to-date list of the known issues of the CAMS reanalysis can be found in the online CAMS documentation website (https://confluence.ecmwf.int/display/CKB/CAMS%3A+Reanalysis+data+documentation). Some of these issues will also be addressed in the future CAMS reanalysis (planned to start production in 2024), including the improvement of the prescribed emission trends, the consistent use of satellite retrieval products and the use of variable $CH_4$ chemical loss rate.

## 5. Summary and conclusions

This technical report documents the first CAMS IFS reanalysis of $CO_2$ and $CH_4$ produced by ECMWF which complements the CAMS reanalysis of reactive gases and aerosols (Inness et al., 2019). The processing chain, assimilated satellite data and underlying model used are described and the resulting reanalysis is evaluated using independent in situ near-surface observations, total column retrievals and in situ atmospheric profile observations. The monthly systematic and random errors of $CO_2$ and $CH_4$ typically range within 1% from 2003 to 2020 with an overall good skill in the representation of synoptic spatial variability and seasonal cycle. The lowest systematic errors occur in the period with maximum number of observations in 2013 and 2014. In 2019 there was a switch from C3S pre-processed satellite products to the near-real time CAMS satellite products because at the time of production the C3S products had not reached 2019. This caused a jump in the quality of the

satellite data and the resulting CAMS GHG reanalysis. For this reason, a new re-run of the CAMS GHG reanalysis from 2019
onwards will be performed with consistent C3S satellite products in the near future.

The comparison of global mean values with observations shows variations in the bias that depend on changes in the assimilated satellite data of around 2 ppm and 10 ppb for $CO_2$ and $CH_4$ respectively, which have the same magnitude as the observed variations in their growth rate. For this reason, we do not recommend the use of this dataset to study changes in the atmospheric
growth rate of $CO_2$ and $CH_4$. Similarly, large biases in stratospheric $CO_2$ and $CH_4$ should also considered when analysing stratospheric signals and trends in the CAMS GHG reanalyses. A list of caveats and limitations that users need to be aware of is provided in Sect. 4.

The slow reduction of the lingering bias in the model background is associated with competing factors at play: (i) the error
growth in the model background associated with the accumulation of systematic errors in emission and natural fluxes; (ii) the limited coverage of observations in time and space (both horizontal and vertical); (iii) the localised impact of observations associated with a short data assimilation window spanning 12 hours.

In order to improve the CAMS reanalysis in future releases we recommend the following actions: (i) increase the number and
coverage of satellite data assimilated from additional satellite missions such as the Copernicus Sentinel-5 Precursor (S5P), Orbiting Carbon Observatory 2 and 3 (OCO-2, https://www.nasa.gov/mission_pages/oco2; OCO-3, https://www.jpl.nasa.gov/missions/orbiting-carbon-observatory-3-oco-3) and Greenhouse gases Observing SATellite-2 (GOSAT-2, https://global.jaxa.jp/projects/sat/gosat2) as well as the latest re-processed satellite products from C3S; (ii) improve the underlying emissions, particularly the extrapolation in near-real time; (iii) couple the chemical loss rate with the
CAMS reanalysis of chemical species (Inness et al. 2019); and (iv) use the IFS inversion capability (McNorton et al., 2022) being developed within the CoCO2 project (coco2-project.eu) for future re-analyses.






**Appendix A**

**Table A.1 List of stations with in situ continuous observations of CO$_2$ and CH$_4$ from GLOBALVIEWplus CO$_2$ ObsPack v6.0 and CH$_4$ ObsPack v1.0 respectively used for the evaluation in Sect. 3.1.**

| Station, Country (site name) | Latitude/Longitude [degrees] | Elevation [m asl] | Data Reference |
|---|---|---|---|
| Alert, Canada (ALT) | 82.45 62.51W | 185 | Worthy et al. (2003) |
| Barrow, AK, USA (BRW) | 71.32N 156.61W | 11 | Peterson et al. (1986) |
| Candle Lake, Canada (CDL) | 53.99N 105.12W | 600 | Worthy et al. (2003) |
| Mace Head, Ireland (MHD) | 53.33N 9.90W | 5 | Ramonet et al. (2010) |
| Fraserdale, Canada (FSD) | 49.88N 81.57W | 210 | Worthy et al. (2003) |
| Kasprowy Wierch, Poland (KAS) | 49.23N 19.98E | 1989 | Rozanski et al. (2003) |
| Schauinsland, Germany (SSL) | 47.92N 7.92E | 1205 | Schmidt et al. (2003) |
| Hegyhatsal, Hungary (HUN) | 46.95N 16.65 | 248 | Haszpra et al (2001) |
| Park Falls, WI, USA (LEF) | 45.95N 90.27W | 472 | Andrews et al. (2014) |
| Puy de Dôme, France (PUY) | 45.77N 2.97E | 1465 | Lopez et al. (2015); Colomb et al. (2020) |
| Argyle, ME, USA (AMT) | 45.03N 68.68W | 53 | Andrews et al. (2014) |
| Sable Island, Canada (WSA) | 43.93N 60.00W | 5 | Worthy et al. (2003) |
| Ryori, Japan (RYO) | 39.03N 141.82E | 260 | Tsutsumi et al. (2005) |
| Moody, TX, USA (WKT) | 31.31N 97.33W | 251 | Andrews et al. (2014) |
| Minamitorishima, Japan (MNM) | 24.28N 153.98E | 8 | Tsutsumi et al. (2005) |
| Yonagunijima, Japan (YON) | 24.47N 123.02E | 30 | Tsutsumi et al. (2005) |
| Tutuila, American Samoa (SMO) | 14.25S 170.56W | 42 | Waterman et al. (1989) |
| Cape Point, South Africa (CPT) | 34.35S 18.49E | 230 | Brunke et al. (2004) |
| Amsterdam Island, France (AMS) | 37.80S 77.54E | 55 | Ramonet et al. (1996) |
| Baring Head Station, New Zealand (BHD) | 41.41S 174.87E | 85 | Stephens et al. (2013) |
| Syowa Station, Antarctica, Japan (SYO) | 69.01S 39.59E | 14 | Schuldt et al. (2020) |
| South Pole, Antarctica, USA (SPO) | 89.98S 24.8W | 2810 | Conway et al. (1990) |



**Table A.2 List of total column stations used for the evaluation in Sect 3.1.**

| Station, country | Latitude/ Longitude [degrees] | Network | Data references |
|---|---|---|---|
| Eureka, Canada | 80.05N 86.42W | TCCON+NDACC | Strong et al. (2019); Batchelor et al. (2009) |
| Ny Ålesund, Norway | 78.9N 11.9E | TCCON+NDACC | Notholt et al., (2019) |
| Thule, Greenland | 76.53N 68.74W | NDACC | Hannigan et al. (2009) |
| Kiruna, Sweden | 67.84N 20.41E | NDACC | Bader et al. (2017) |
| Sodankylä, Finland | 67.37N 26.63E | TCCON+NDACC | Kivi et al. (2014); Sha et al. (2021) |
| Harestua, Norway | 60.2N 10.8E | NDACC | De Mazière et al. (2018) |
| St Petersburg, Russia | 59.90N 29.80E | NDACC | Makarova et al. (2015) |
| East Trout Lake, Canada | 54.35N 104.99W | TCCON | Wunch et al. (2018) |
| Bialystok, Poland | 53.23N 23.02E | TCCON | Deutscher et al. (2015) |
| Bremen, Germany | 53.1N 8.85E | TCCON | Notholt et al. (2014) |
| Karlsruhe, Germany | 49.1N 8.44E | TCCON | Hase et al. (2015) |
| Paris, France | 48.85N 2.36E | TCCON | Te et al. (2014) |
| Orléans, France | 47.97N 2.11E | TCCON | Warneke et al. (2014) |
| Garmisch, Germany | 47.48N 11.06E | TCCON+NDACC | Sussmann and Rettinger (2018a); Sussmann et al. (2012); Hausmann al. (2016) |



| Zugspitze, Germany | 47.42N 10.98E | TCCON+NDACC | Sussmann and Rettinger (2018b) |
|---|---|---|---|
| Jungfraujoch, Switzerland | 46.55N 7.98E | NDACC | Zander et al. (2008) |
| Park Falls, WI, USA | 45.94N 90.27W | TCCON | Wennberg et al. (2017) |
| Rikubetsu, Japan | 43.46N 143.77E | TCCON+NDACC | Morino et al. (2016); De Mazière et al. (2018) |
| Boulder, CO, USA | 39.99N 105.26W | NDACC | Ortega et al. (2021) |
| Indianapolis, IN, USA | 39.86N 86W | TCCON | Iraci et al. (2016) |
| Four Corners, USA | 36.8N 108.48W | TCCON | Dubey et al. (2014) |
| Lamont, OK, USA | 36.5N 97.49W | TCCON | Wennberg et al. (2016) |
| Anmeyondo, South Korea | 36.54N 126.33E | TCCON | Goo et al. (2014) |
| Tsukuba, Japan | 36.05N 140.12E | TCCON | Morino et al (2016) |
| Nicosia, Cyprus | 35.14N 33.38E | TCCON | Petri et al. (2020) |
| Edwards, CA, USA | 34.96N 117.88W | TCCON | Iraci et al. (2016) |
| JPL, CA, USA | 34.2N 118.18W | TCCON | Wennberg et al. (2016) |
| Pasadena Caltech, CA, USA | 34.14N 118.13W | TCCON | Wennberg et al. (2015) |
| Saga, Japan | 33.24N 130.29E | TCCON | Kawakami et al. (2014) |
| Heifei, China | 31.9 N 117.17E | TCCON | Liu et al. (2018) |



| Izaña, Spain | 28.3N 16.48W | TCCON+NDACC | Blumenstock et al. (2014); García et al. (2021) |
|---|---|---|---|
| Mauna Loa, HI, United States | 19.54N 155.58W | NDACC | Hannigan et al. (2009 |
| Altzomoni, Mexico | 19.12N 98.66W | NDACC | De Mazière et al. (2018); |
| Burgos, Philippines | 18.53N 120.65E | TCCON | Morino et al. (2018) |
| Manaus, Brazil | 3.21S 60.6W | TCCON | Dubey et al. (2014) |
| Ascension Island, UK | 7.92S 14.33W | TCCON | Feist et al. (2014) |
| Darwin, Australia | 12.43S 130.89E | TCCON | Griffith et al. (2014) |
| Reunion St Denis, France | 20.9S 55.49E | TCCON+NDACC | De Mazière et al. (2014) |
| Reunion Island, Maido, France | 21.1S 55.4E | NDACC | Zhou et al. (2018) |
| Wollongong, Australia | 34.41S 150.88E | TCCON+NDACC | Griffith et al. (2014); De Mazière et al. (2018); |
| Lauder, New Zealand | 45.05S 168.68E | TCCON+NDACC | Sherlock et al. (2014a, 2014b); Pollard et al. (2019) Bader et al. (2017) ; Pollard et al. (2017) |
| Arrival Heights, Antarctica | 77.83S 166.67E | NDACC | Bader et al. (2017) |




**Table A.3 List of AirCore sites (from NOAA_AirCore_data_v20210813, Baier et al., 2021) used for the evaluation in Sect. 3.2.**

| Site, country | Latitude /Longitude [degrees] |
|---|---|
| Boulder, CO, USA | 40.03N 103.74W |
| Lamont, OK, USA | 36.85N 98.21W |
| Lauder, New Zealand | 45.50S 169.47E |
| Sodankylä, Finland | 67.41N 26.31E |
| Park Falls, WI, USA | 45.97N 90.32W |
| Edwards, CA, USA | 34.65N 117.29W |
| Traînou, France | 48.48N 1.16E |

**Code and data availability**

The IFS forecasting and reanalysis system is not for public use as the ECMWF Member States are the proprietary owners. The
resulting dataset is however freely available on the Copernicus Atmosphere Data Store. The CAMS GHG reanalysis can be
accessed through the CAMS Atmosphere Data Store (ADS) at https://doi.org/10.24380/8fck-9w87.  The format is available in
both GRIB and NetCDF. The data record starts on 1 January 2003 00UTC and currently stops on 31 December 2020. Recent
months will be added over time as soon as the reanalysis procedure and its validation are completed. The original data was
available either as spectral coefficients with a triangular truncation of T255 or on a reduced Gaussian grid with a resolution of
N128. But for the ease of the user, fields were interpolated from their native representation to a regular 0.75°x0.75° latitude
longitude grid. For sub-daily data for the CAMS reanalysis is archived with a 3-hourly time step (0, 3, 6, 9, 12, 15, 18, 21
UTC). Pre-computed monthly averages are also directly available for all fields. The 3D fields are available on two different
vertical resolution: 25 pressure levels (1000, 950, 925, 900, 850, 800, 700, 600, 500, 400, 300, 250, 200, 150, 100, 70, 50, 30,
20, 10, 7,5, 3, 2, 1 hPa) and 60 σ-hybrid model levels which are described
at https://www.ecmwf.int/en/forecasts/documentation-and-support/60-model-levels. The data records have 18 2D radiation
fields, 2 vertically integrated atmospheric content of $CO_2$ and $CH_4$ (column-mean mole fractions, 14 2D surface fluxes
variables, 32 2D meteorological fields and 16 3D fields including meteorological and greenhouse gases fields. A complete
listing of the variables included in the CAMS GHG reanalysis is provided in the ADS
(https://ads.atmosphere.copernicus.eu/cdsapp#!/home).




**Acknowledgements**

The Copernicus Atmosphere Monitoring Service is operated by the European Centre for Medium-Range Weather Forecasts on behalf of the European Commission as part of the Copernicus Programme (http://copernicus.eu). The satellite data assimilated in the CAMS GHG reanalysis were obtained from the ESA GHG-CCI project
(https://climate.esa.int/en/projects/ghgs/Data) and the Copernicus Climate Change Service (C3S) Climate Data Store (https://cds.climate.copernicus.eu). The data used to evaluate the CAMS GHG reanalysis were obtained from: the Total Carbon Column Observing Network (TCCON) Data Archive hosted by CaltechDATA at https://tccondata.org.; $CH_4$ from the Network for the Detection of Atmospheric Composition Change (NDACC, www.ndacc.org; Sussmann et al., 2011; 2013; De Mazière et al., 2018) (see Table A2); the Observation Package (ObsPack) GLOBALVIEWplus Data Products
https://gml.noaa.gov/ccgg/obspack; the NOAA AirCore dataset https://gml.noaa.gov/ccgg/arc/?id=144; and the NOAA Greenhouse Gas Marine Boundary Layer Reference https://gml.noaa.gov/ccgg/mbl/mbl.html.

For the NDACC data, the National Center for Atmospheric Research is sponsored by the National Science Foundation. The NCAR FTS observation programs at Thule, GR, Boulder, CO and Mauna Loa, HI are supported under contract by the
National Aeronautics and Space Administration (NASA).  The Thule work is also supported by the NSF Office of Polar Programs (OPP).  We wish to thank the Danish Meteorological Institute for support at the Thule site and NOAA for support of the MLO site. The multi-decadal monitoring program of University of Liège at the Jungfraujoch station has been primarily supported by the F.R.S.-FNRS and BELSPO (both in Brussels, Belgium) and by the GAW-CH programme of MeteoSwiss. The International Foundation High Altitude Research Stations Jungfraujoch and Gornergrat (HFSJG, Bern)
supported the facilities needed to perform the FTIR observations at Jungfraujoch.

The authors are also grateful to László Haszpra for his comments on the Hegyhátsál tall-tower station, Xin Lan and Ed Duglokencky for the feedback on the use of NOAA/GML MBL data; Debra Wunch for providing insight into the potential impact of polar vortex on the XCH4 data at East Trout Lake TCCON site.

**Author contributions**

Anna Agustí-Panareda, Jérôme Barré and Sébastien Massart prepared the manuscript with the rest of the co-authors, monitored the simulation, testing the impact of the surface fluxes and assimilated observations, as well as fixing bugs. Anna Agustí-Panareda was responsible of the forecasting model implementation and performed the evaluation of the vertical profiles and trends. Antje Inness led the CAMS reanalysis work and was responsible of running and monitoring the reanalysis simulations.
Sébastien Massart developed the data assimilation of satellite products. Mark Parrington provided support on the biomass burning emissions and postprocessed the fire emission data in Fig. 3. Mel Ades worked on documenting the data assimilation aspects of the GHG reanalysis. Johannes Flemming, Zak Kipling and Luca Cantarello provided technical support for the diagnostics, bug fixes implemented during the testing stage and testing of the CAMS NRT satellite observations. Luke Jones provided technical support with the mass interpolation scheme and diagnostic tools. Miha Ratzinger was responsible for post-
processing the dataset into the Copernicus Atmosphere Data Store. Roberto Ribas and Martin Suttie performed the satellite data acquisition and pre-processing. Richard Engelen and Vincent-Henri Peuch coordinated the efforts on the CAMS reanalyses. Bavo Langerock, Henk Eskes, Michel Ramonet and Jérôme Tarniewicz were responsible of validating the reanalysis dataset with independent observations including the production of surface and column $CO_2$ and $CH_4$ evaluation plots. Ilse Aben, Tobias Borsdorff, Michael Buchwitz, Cyril Crevoisier, Otto Hasekamp, Nicolas Meilhac, Stefan Noel,



Maximilian Reuter and Lianghai Wu provided support for the use of the satellite retrieval products. All authors reviewed and edited the manuscript.

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
