# Peer review of "Technical note: The CAMS greenhouse gas reanalysis from 2003 to 2020"

_EGUsphere, 2022_

## Author Comment (AC1)

**Technical note: The CAMS greenhouse gas reanalysis from 2003 to 2020**

**Reply to reviewer 1**

We are very grateful for the insightful comments from the reviewer to improve the documentation of the CAMS greenhouse gas reanalysis and in the manuscript and the suggestions for potential improvements to be considered in future reanalyses. We have addressed the different comments of the reviewer (in black) to clarify the different aspects of the reanalysis including the selection/quality of observations assimilation, the constrain of the annual global growth rates and the quality of the prescribed surface fluxes (see text in blue).

Review of "The CAMS greenhouse gas reanalysis from 2003 to 2020" by Agusti-Panareda et al.

This paper describes the greenhouse gases ($CO_2$ and $CH_4$) reanalysis as prepared for the Copernicus Atmosphere Monitoring Service (CAMS). Manuscript draft is well prepared for understanding of the work that is carried out. The method is well described but the results are not fully satisfactory yet. Many traditional inversions produced better statistics for model simulated $CO_2$ and $CH_4$ concentrations by optimising the surface sources and sinks (e.g., Chandra et al., ACP 2022). However, the approach holds different promises for 4D concentration product of $CO_2$ and $CH_4$ in near real time along with other meteorological parameters of ECMWF weather forecasting system. In general I recommend publication of the manuscript as "Technical note", after accounting for some of my comments below.

Line 48: are the SCIAMACHY and IASI data good enough for assimilation ? the growth rate may be fine but the spatial distribution are probably of poor quality.

I have found later that this is mentioned in the caveats. No action is needed but I am just making sure to mention the points of concern.

The rationale for the selection of the observations assimilated in the CAMS reanalysis and their quality will be emphasized in the revised version of the manuscript.

Figure 1: I have several doubts on these plots.

1) the total column $CO_2$ values on Dec 2020 are of similar magnitude of $CO_2$ at MLO (414 pm as per SIO flask data). I was thinking that global mean $XCO_2$ will be a couple ppm lower than MLO value. Please confirm.

Figure 1 gives an overview of the CAMS GHG reanalysis dataset including global trends and spatial/temporal distribution at seasonal sales. The global mean has an error ranging from -0.7 to 3.5 ppm. Figure 14 shows that global mean error in 2020 is +1.35

ppm. This is consistent with the assessment of the reviewer. The problem of constraining the global mean and annual growth rate is documented in section 3.3 and mentioned in section 4 (point 6). The range of error in the global mean will be added in the revised version of the manuscript.

2) What is the advantage of showing 2003-2020 mean as opposed to just 2020 seasonal means?

The purpose here is to show the typical seasonal cycle. For this reason, using 2003-2020 mean as opposed to just 2020 seasonal means is better because individual years can be affected by the large inter-annual variability of biogenic fluxes (e.g. during el Nino years). This will be mentioned in the revised manuscript.

Figure 2: Should the two-way arrows between Forecast "Sphere" and Surface flxes "Maps" be one-way? From Fluxes to Forecast

The two-way arrows indicate the two-way coupling between the atmospheric model and the model of biogenic surface fluxes. The surface fluxes are affected by the forecast of temperature, radiation, humidity and soil moisture and the atmospheric CO2 in the forecast is affected by the surface biogenic fluxes. This will be clarified in the revised version of the manuscript.

Table 2: You could use FF-CO2 emissions EDGARv6.0 etc. or the GridFED by UEA group. Similarly many of the flux components of CO2 and CH4 should be revised by using the most recent flux data sources. I not saying for this paper but in the near future.

The current reanalysis cannot be changed as the reanalysis system requires a consistent configuration of the model and prescribed input fluxes throughout the whole reanalysis, but we will certainly use a newer version of the anthropogenic emissions in the next CAMS reanalysis planned for 2024. This will be mentioned in the revised manuscript.

Line 352: I think it would have been nice to add/subtract some CO2 flux to make the CO2 flux budget consistent with observed CO2 growth rate. At the very least the land biosphere + ocean exchange could be added/subtracted with an offset to balance all sources+sinks = 2.12 * Growth rate. Then the statistics may be look that bad.

This adjustment of the fluxes with the CO2 growth rate will be considered in the next CAMS re-analysis. There is the caveat that the growth rate will have to be adjusted in near-real time which might not be feasible. A note of this option will be made in the discussion section of the revised manuscript.

Figure 7: It would be nicer to use a discrete colour scale (like in Fig. 7). Otherwise, it is hard to distinguish between medium blue/red from dark blue/red

Also consider showing a Taylor diagram, like I see below for the TCCON sites

We will change Figure 7 to use a discrete colour scale and add a Taylor diagram in the revised manuscript.

Do you need this Figure 14 ? If needed I suggest a more detailed site information is shown before Fig. 7. The Surface, MBL, TCCON, NDAAC, AirCore etc. sites can be shown in different symbols

The map of the MBL sites is useful to be able to interpret the observed global mean produced by NOAA. A detailed list of TCCON, The NDACC, AirCore is provided in the Appendix. In the revised manuscript the map in Figure 14 will be updated to show the different types of observations with different symbols and placed before Fig. 7 as suggested by the reviewer.

---

## Author Comment (AC4)

**Technical note: The CAMS greenhouse gas reanalysis from 2003 to 2020**

**Reply to reviewer 2**

We are very grateful for the insightful comments from the reviewer to improve the documentation of the CAMS greenhouse gas reanalysis for potential users. We have addressed the comments of the reviewer (in bold) to clarify the different aspects of the reanalysis including the selection/quality of the assimilated observations, the different model components and the characteristics of the prescribed surface fluxes.

**Review of "The CAMS greenhouse gas reanalysis from 2003 to 2020" by Agusti-Panareda et al.**

**This article presents a description of a new reanalysis dataset of greenhouse gases (GHGs). This is the first GHG reanalysis dataset ever produced using data assimilation techniques that adjust GHG concentrations rather than GHG fluxes. As such, it is a formidable achievement but at the same time, there is considerable room for improvement (as noted by the authors in sections 4 and 5). Reanalyses are typically improved after feedback from the user community and/or with improved data assimilation techniques or increased numbers of observations. Thus an important aspect of reanalysis production is the dissemination and documentation of the products to encourage and inform the potential user community. That is the main purpose of this article. As such, the comments below are primarily aimed at improving the description of the product to potential users.**

**Specific comments**

1. **Section 2.2 and Table 1: It would be useful to know how the satellite instruments were selected. For example, OCO-2 was not used. In section 5, it is mentioned that it will be used in future versions, but it is useful for the user to understand the rationale behind the selection of instruments for this product.**

   The main rationale for the selection of the data was the operational availability of NRT data, as the re-analysis is foreseen to catch up and run in near-real time eventually. Currently there is no near-real time product of the OCO-2 data and that is why OCO-2 has not been used in the current CAMS near-real time products (including re-analysis). We are working with LSCE to have a near-real time OCO-2 product based on a neural network retrieval (Breon et al., 2022; https://amt.copernicus.org/articles/15/5219/2022/amt-15-5219-2022-discussion.html). This will allow us to use OCO-2 in the next CAMS re-analysis. The rationale for the observation data selection will be clarified in the revised manuscript.

2. **Line 99 and section 2.3: Figure 2 shows 2-way interaction between the forecast and surface fluxes. Please explain how the forecast influences the surface fluxes in section 2.3 or line 99.**

The two-way arrows indicate the two-way coupling between the atmospheric forecast and the biogenic surface fluxes. The surface fluxes are affected by the forecast of temperature, humidity, radiation and soil moisture and the atmospheric $CO_2$ in the forecast is affected by the surface biogenic fluxes. This will be clarified in the revised version of the manuscript.

3. **Section 2.3: There is no mention of Figure 3 in the text. It is useful to understand why this Figure is presented, and the main message behind it.**

Figure 3 shows the seasonal, inter-annual variability and trend of the surface fluxes. This is important because one of the caveats of the re-analysis is that $CH_4$ emissions are kept fixed from 2010 onwards, while for $CO_2$ anthropogenic fluxes we apply an extrapolation, and the biogenic fluxes are modelled and therefore have an inter-annual variability. In the revised manuscript, Figure 3 is mentioned and referred to when explaining the differences between different fluxes in terms of inter-annual variability and seasonal cycles.

4. **Line 184: What is the rationale behind the choice of EDGAR versus higher spatial resolution datasets for anthropogenic emissions?**

EDGAR produces global emissions for both $CO_2$ and $CH_4$ at a relatively high resolution of 0.1 degrees (compared to 80km resolution of the CAMS re-analysis). The problem with EDGAR is the extension to NRT and the caveats associated with large inter-annual variability during the COVID period. This will be clarified in the revised manuscript.

5. **Line 217: Typo: Tl255 should be TL255 presumably.**

This will be corrected in the revised manuscript.

6. **Line 302: I do not see the 20 ppm error in Figure 5. If the vertical scale in Fig. 5c (bottom panel) is linear then the green and red dashed curves seem to overlap after 2019, meaning errors of less than 5 ppm.**

Yes, indeed. There is a typo here, 20ppm should be 2ppm. The sentence "accompanied by a large standard deviation error (~20ppm, cf. dashed lines in same panel and figure)" has been removed as the degradation associated with IASI is mainly detected from the large step change in the averages of the departures.

7. **Lines 368-370: Taylor diagrams of Figs. 8b, 11b. What is the normalization used on the radial axes? Presumably it is the observed standard deviation for a given site. Why was this normalization needed? Standard Taylor diagrams do not do a normalization. Presumably it allows for better comparison among sites with very different observed variability.**

The reviewer is correct. Assuming we have 20 sites, then there are 20 time series for the model and the observed NDACC ground based data (gb), each with its own variability. In order to plot them together, the standard deviation of the model time series is used to normalize the timeseries (i.e for a site, the time series model or gb is divided with the std of the model time series ): that is why the model time series is always at 1 for each site. The correlations are independent of such a normalization and from the location of a site in the Taylor plot you can deduce if the model has higher variability (the site is plotted with a distance <1 to the origin) or lower (the site is plotted at a distance >1 from the origin) compared to the variability in the gb data. This will be clarified in the revised manuscript.

8. **Lines 457-8 and Fig. 13: How was the site for Fig. 13 chosen? Is it a typical example, or a good example? It is nice to see that the overall structure of the profile (esp. for $CO_2$) is well captured in boreal summer in France. It would be interesting to see the comparison in the southern hemisphere. Are the general biases of Fig. 12 more evident in New Zealand?**

The site of Traînou (France) has been used as a good example of the synoptic variability during the summer, more than the systematic errors. As the reviewer suggested, we have plotted the only profile available from Lauder (New Zealand) (see Figure S1 in supplement) which shows much larger errors in September, particularly near the surface and in the stratosphere. In the revised manuscript we will add an extra figure as a Supplement to illustrate the errors at different sites across different latitude bands.

[Figure]

[Figure]

*Figure S1. Vertical mole fraction profiles of $CO_2$ and $CH_4$ from the CAMS GHG reanalysis (dash line) and AirCore observations (solid line) at Lauder (New Zealand, see Table A3) over the period in June 2019.*

9. **Line 496: Since this is a reanalysis, why were the in situ data not assimilated?  Would they be enough to better constrain the global growth rate?  Will this be done in the future?**

   The CAMS reanalysis and generally the reanalysis at ECMWF aim to eventually run in near-real time. Since the availability of quality controlled in situ data is not generally available in near real time and we use a global model with a short window data assimilation (i.e. 12 hours) that adjusts only the GHG concentrations (i.e. not the fluxes), the impact of the sparse in situ data would be small as the increments are very localised in time (i.e at the beginning of the data assimilation window) and space (around the station near the surface). Preliminary tests showed that the increments around the in situ stations are dispersed by the atmospheric transport in less than 12 hours. In the future, we will explore the possibility of using in situ data with the inversion capability currently being developed in the CoCO2 project. The flux adjustment will result in a longer-lived impact of the impact of the in situ observations, as we expect the resulting atmospheric CO2 and CH4 corrections will not be as localised in time and space, particularly when using a longer data assimilation window with the inversion capability. All these aspects related to the future potential assimilation of in situ data will be clarified in the revised version of the manuscript.

---

## Author Response (AR2)

**Technical note: The CAMS greenhouse gas reanalysis from 2003 to 2020**

Many thanks to the reviewer for pointing out the missing updates in Figures 8 and 10 (upper panels have now been updated with the correct discrete colour bars). The labelling of the last figures has also been corrected to Figure 15.